# Does Egalitarian Fairness Lead to Instability? The Fairness Bounds in Stable Federated Learning Under Altruistic Behaviors

**Jiashi Gao**[1], **Ziwei Wang**[1,2], **Xiangyu Zhao**[3], **Xin Yao**[4], **Xuetao Wei**[1]*

[1]Southern University of Science and Technology
[2]University of Birmingham
[3]City University of Hong Kong
[4]Lingnan University
{12131101,12250053}@mail.sustech.edu.cn
xy.zhao@cityu.edu.hk
xinyao@ln.edu.hk
weixt@sustech.edu.cn

## Abstract

Federated learning (FL) offers a machine learning paradigm that protects privacy, allowing multiple clients to collaboratively train a global model while only accessing their local data. Recent research in FL has increasingly focused on improving the uniformity of model performance across clients, a fairness principle known as egalitarian fairness. However, achieving egalitarian fairness in FL may sacrifice the model performance for data-rich clients to benefit those with less data. This trade-off raises concerns about the stability of FL, as data-rich clients may opt to leave the current coalition and join another that is more closely aligned with its expected high performance. In this context, our work rigorously addresses the critical concern: *Does egalitarian fairness lead to instability?* Drawing from game theory and social choice theory, we initially characterize fair FL systems as altruism coalition formation games (ACFGs) and reveal that the instability issues emerging from the pursuit of egalitarian fairness are significantly related to the clients' altruism within the coalition and the configuration of the friends-relationship networks among the clients. Then, we theoretically propose the optimal egalitarian fairness bounds that an FL coalition can achieve while maintaining core stability under various types of altruistic behaviors. The theoretical contributions clarify the quantitative relationships between achievable egalitarian fairness and the disparities in the sizes of local datasets, disproving the misconception that egalitarian fairness inevitably leads to instability. Finally, we conduct experiments to evaluate the consistency of our theoretically derived egalitarian fairness bounds with the empirically achieved egalitarian fairness in fair FL settings.

## 1 Introduction

Federated learning (FL) has emerged as a significant learning paradigm in which clients utilize their local data to train a global model collaboratively without sharing data and has attracted researchers from various fields, especially in domains where data privacy and security are critical, such as healthcare, finance, and social networks [1, 2, 3]. In vanilla FLs, different clients sharing the same global model may experience varying performance (i.e., error) due to differences in the amount

---

*Corresponding author

38th Conference on Neural Information Processing Systems (NeurIPS 2024).

of data contributed by each client. As clients may lack access to significant data resources due to objective or unavoidable factors, such as regional specificity and historical inequalities, recent works [4, 5, 6, 7, 8, 9] have focused on ensuring that the performance of global model across the clients roughly comparable or even equal, termed as "egalitarian fairness". In FLs complying with egalitarian fairness, the performance of the global model on clients contributing high data resources may decrease to enhance the performance on other clients contributing lower data resources, particularly when local datasets are heterogeneous. This case could potentially cause higher-resource clients to leave the current grand coalition and form a sub-coalition to achieve desired performance, thereby disrupting the core stability [10, 11] of the FL. In game theory, a coalition is said to be core-stable if no sub-coalition exists such that the players of that sub-coalition can obtain a payoff no less than they are allocated in the original coalition and at least one player of that sub-coalition can obtain a payoff greater than it was allocated in the original coalition. Otherwise, the original coalition is said to be blocked.

Prior works [12, 13, 14, 15] have primarily examined the effects of pursuing performance optimality on the core stability of FLs, leaving the effect of fairness unexplored. These works view the FL process as a hedonic game [16] and assume that clients are selfish, with clients' utility functions being their own performance and only influenced by other individuals in the same coalition. However, in the real world, clients' behaviors are more complex: clients may form friends-relationships and exhibit altruism towards each other. Various external factors, such as commercial ties, welfare purpose, or shared research interests, etc, influence the relationships between clients. For instance, branches of the same corporation can be considered as *friend*s due to common commercial goals. In contrast, branches from rival companies may be viewed as *non-friend*s. Hospitals collaborating on developing a diagnostic model for welfare purposes can also be regarded as *friend*s. Under altruistic behavior, a player's utility depends not only on their own performance but also on the performance of their friends. Furthermore, altruistic behaviors in FL, which have been largely overlooked in previous work, are particularly crucial when considering the impact of egalitarian fairness on stability. Altruistic behaviors influence the willingness of clients to sacrifice a certain degree from optimal performance [17, 18]. Considering the friend-relationships, Nguyen et al. [19] introduced altruistic hedonic games (AHGs), wherein players gain utility from their own valuation of the current coalition and the valuation of their friends within the same coalition. However, players who are friends may not be in the same coalition, but they can still influence each other's utility in the coalitions. For instance, altruistic hospital clients might choose to leave one coalition to join another where their friend hospitals are, aiming to enhance each other's performance collaboratively. When the FL process is described as AHGs, which excludes some friends' influence on a player's altruistic behavior, it may lead to a core-stable coalition in terms of local utilities but non-Pareto optimal in local errors. An alternative coalition structure exists where each client encounters an error not higher than that observed within the core-stable coalition structure, with at least one client experiencing a lower error.

In this context, we pose the following open questions: *How does egalitarian fairness affect the stability of FLs? How does this impact vary when clients exhibit altruistic behaviors? What is the optimal egalitarian fairness that a stable FL can achieve?*

**Our contributions.** We initiate discussions to address the above questions by proposing the game model under altruistic behaviors in Section 4. Based on the degree of altruism, we define three types of client behavior: purely selfish (non-altruistic), purely altruistic, and friendly altruistic. Existing research on egalitarian fairness mainly involves two implementations: optimizing the global model towards the worst-performing client [5, 20, 6] or simultaneously to all clients [4, 7, 8]. Correspondingly, we further distinguish altruism into a min-based and a mean-based aggregation of the friends' performance: welfare altruism and equal altruism. We model the collaborative training involving clients with altruistic behavior as altruism coalition formation games (ACFGs), where clients' utilities depend on a network of friends via one of the above cases of altruism. We theoretically demonstrate that the ACFGs result in a core-stable grand coalition, which is also Pareto-optimal in local errors.

In Section 5, we demonstrate in detail that egalitarian fairness does not necessarily lead to instability; rather, it depends on the behaviors of the clients and the topology of the friends-relationship networks. We theoretically propose the optimal egalitarian fairness bounds that a core-stable FL can obtain. These bounds are functions of the disparities between the dataset sizes within the clients' friends network and the global minimal one. The established fairness bounds can be utilized to set a reasonable level of fairness in fair FLs. The proofs for these fairness bounds are modular. Beyond

establishing tight fairness bounds, part of the contribution of this work lies in the analysis methodology used to derive these bounds.

The main contributions in this paper are primarily theoretical. However, we also conduct experiments to assess the consistency of the theoretically proposed fairness bounds with the empirically achieved fairness in fair FLs.

## 2    Related work

**Stability in FL.** Existing results on stability in FL mainly focus on a particular aspect of the entire issue. For instance, Donahue et al. [12] have first formulated the model sharing problem as a hedonic game, in which each player derives some cost (error) from the coalition they join. The aim of their research is to identify the conditions when the grand coalition or partitions of players are stable for varying federation aggregation mechanisms. In their subsequent work [15], they have further contributed to calculating an optimal (minimum error) federating arrangement and analyzing the differences between stable arrangements and optimal arrangements using the canonical game-theoretic tools of the Price of Anarchy and the Price of Stability. Chaudhury et al. [14] have defined a fairness concept based on core stability, which requires that no subset of agents can benefit significantly than in grand coalition, and proposed CoreFed, an FL protocol to implement a core-stable predictor. Blum et al. [21] have proposed a stable and envy-free equilibrium-based collaboration protocol to meet the client's learning objectives while keeping their local data collection burden low. However, the impact of fairness on FL stability remains unexplored in existing work. Moreover, current research on the stability of FL is limited to the assumption of selfish individuals who compete with each other to win the game and maximize their own profits.

**Altruism in game theory.** From more background on game theory and social choice theory, altruism [22, 23, 24] has been considered for cooperative games to date, in which the players' payoffs in the resulting game do not only depend on their individual payoffs but also on the neighborhood graph and the aggregation functions that reflect the social context. Numerous studies [25, 26, 27, 28] have highlighted the ubiquity and rationality of human altruism from economic, neural, and evolutionary perspectives. Rothe [22] has systematically introduced certain notions of altruism into existing game-theoretic models in both non-cooperative and cooperative games. Kerkmann et al. [29] have formally distinguished three degrees of altruism: selfish-first, equal-treatment, and altruistic-treatment preferences and studied both the axiomatic properties and the computational complexity of stability in altruistic hedonic games. In their other work [30], they have additionally studied stability notions and computational analysis of altruism in coalition formation games. These works in game theory, where altruistic players only prefer coalitions with more friends, are constrained in the FL setting when additional model performance on both the player and their friends need to be considered.

## 3    Preliminaries

**Model and assumptions.** Let's consider a setup involving $N$ clients, where the $i$-client possesses a local dataset $\mathcal{D}_i$ of size $n_i$. The local datasets are samples drawn from data distribution $P_i(y|x)$. Each client can train a model $\mathcal{M}$ with parameter $\theta_i$ locally, resulting in $P(\mathcal{M}(x, \theta_i)|x) \rightarrow P_i(y|x)$. In addition to local training, clients can participate in collaborative training with other clients and form an FL coalition, denoted as $\pi$. Within this coalition, clients share their local model parameters. These shared parameters are then aggregated according to a specific rule, $\theta = \frac{\sum_{i \in \pi}(n_i \cdot \theta_i)}{\sum_{i \in \pi} n_i}$.

The above weighted aggregation known as FedAvg [31] is a widely used FL aggregation mechanism. We measure the aggregated global model's performance on the client's local dataset in the coalition through the expected error $err_i(\pi) = \mathbb{E}_{x,y \sim \mathcal{D}_i}[\|\mathcal{M}(x, \theta) - y\|]$. A lower $err_i(\pi)$ implies that the coalition $\pi$ is more valuable for the $i$-th client. To determine a tight fairness bound, an FL model that gives exact errors for each client is necessary. Therefore, we build our work on the mean estimation task developed by Donahue et al. [12], which develops the closed-form local errors as in Lemma 1. It's important to note that while we utilize this model, the questions posed by Donahue et al. [12] significantly differ from the focus of this paper: they focused on developing the error models in FL setting and identifying the stable conditions for selfish-clients forming coalitions, while our work explores the relations between fairness and core-stability in FL settings when clients exhibit altruistic behaviors.

**Lemma 1** *(Corollary 4.3 in [12]) In an FL setting with $N$ clients, each client possesses a local dataset $\mathcal{D}_i$ of size $n_i$. The local dataset of each client $\mathcal{D}_i$ is with mean $\theta_i$ and standard deviation $\epsilon_i$, where $(\theta_i, \epsilon_i^2) \sim \Theta$. When FL trains a global model for mean estimation and employs FedAvg for aggregation, the expected mean squared error (MSE) for a client with $n_i$ samples within coalition $\pi$ is as follows,*

$$err_i(\pi) = \frac{\mu_e}{\sum_{j \in \pi} n_j} + \sigma^2 \cdot \frac{\sum_{j \in \pi, j \neq i} n_j^2 + \left(\sum_{j \in \pi, j \neq i} n_j\right)^2}{\left(\sum_{j \in \pi} n_j\right)^2}, \tag{1}$$

*where $\mu_e = \mathbb{E}_{(\theta_i, \epsilon_i^2) \sim \Theta}\left[\epsilon_i^2\right]$ denotes the expected value of the variance of the dataset distribution, and $\sigma^2 = var(\theta_i)$ denotes the variance between the means of the clients' local datasets.*

**Fairness definition.** We explore egalitarian fairness in Definition 1, a fairness notion that assesses whether the global model exhibits equitable performance across clients with varying data resources.

**Definition 1** *(Egalitarian fairness) For the clients within a coalition $\pi$ holding datasets of varying sizes $\{n_1, n_2, ..., n_N\}$ and experiencing errors $\{err_1(\pi), err_2(\pi), ..., err_N(\pi)\}$, the coalition structure $\pi$ satisfy $\lambda$-egalitarian fairness if there exists a constant $\lambda$ such that,*

$$\frac{err_i(\pi)}{err_j(\pi)} \geq \lambda, n_i \leq n_j. \tag{2}$$

Here, $\lambda$ is the fairness bound. When $\lambda = 1$, the coalition $\pi$ is said to satisfy strict egalitarian fairness.

## 4 Game model

Before delving into the impact of fairness on the stability of the FL, we first define the concepts from the perspective of game theory and social choice theory. All proofs for this section are in Appendix.

**Definition 2** *(Value) In the context of collaborative gaming, the value quantifies the payoff accrued to the $i$-th player as a result of participating within the current coalition $\pi$. Within the framework of FL, the value is defined as the error of the global model evaluated on the $i$-th client's local dataset as $v_i(\pi) = err_i(\pi)$.*

**Definition 3** *(Friend) In a broader sociological context, the friend is considered the most intimate, trustful, and voluntarily chosen tie people maintain. Within the framework of FL, the friend set of the $i$-th client, denoted as $F_i$, is defined as the clients whose value is also expected to be better when $i$-th client makes a coalition participation decision.*

**Definition 4** *(Core stability) The grand coalition $\pi_g$ (the coalition consisting of all players) is considered to be core-stable if there does not exist nonempty sub-coalition $\pi_s \subset \pi_g$ such that $\pi_s \succ_i \pi_g$ for $\forall i \in \pi_s$, where $\succ$ is used to denote a preference relation. In other words, no nonempty sub-coalition $\pi_s \subset \pi_g$ blocks $\pi_g$.*

Based on the above notions, we define *altruism* in FL as the client's choice to remain within or exit the existing coalition $\pi$, influenced by the acquired values of their friends $\forall f \in F_i$. Previous research on egalitarian fairness has primarily focused on two approaches: optimizing the global model towards the worst-performing client [5] or simultaneously to all clients [6]. Correspondingly, we classify altruism into two categories: *welfare altruism* and *equal altruism*. In the case of *welfare altruism*, a client's primary concern is for the friend who is in the worst situation. On the other hand, *equal altruism* implies that a client shows equal concern for all friends. The aggregate values for *welfare* and *equal altruism* are defined as $\min_{f \in F_i}(\{v_f(\pi)\})$ and $\frac{1}{|F_i|} \sum_{f \in F_i} v_f(\pi)$, respectively.

### 4.1 Client behavior

We divide client behavior into three types based on the degree of altruism: *purely selfish (non-altruistic)*, *purely altruistic* and *friendly altruistic*.

• **Purely selfish**: A *purely selfish* client opts to stay or leave the current coalition $\pi$, based solely on its own value, i.e., according to value definition, a *purely selfish* client prefers to join a coalition that offers a lower value $v_i$ and conversely, to leave a coalition that results in a higher value $v_i$. The utility function of a *purely selfish* client is defined as $u_i^{ps}(\pi) = v_i(\pi)$.

- **Purely altruistic**: A *purely altruistic* client chooses to stay or leave the current coalition $\pi$, only based on the value received by its friends. Corresponding to *welfare* and *equal altruism*, the utilities of a *purely altruistic* client are defined as $u_i^{pa}(\pi) = \max_{f \in F_i}(\{v_f(\pi)\})$ and $u_i^{pa}(\pi) = \frac{1}{|F_i|}\sum_{f \in F_i} v_f(\pi)$, respectively.

- **Friendly altruistic**: A *friendly altruistic* client decides whether to stay or leave the current coalition, $\pi$, based on its own value and the value its friends receive. Corresponding to *welfare* and *equal altruism*, the utility function of a *friendly altruistic* client is defined as $u_i^{fa}(\pi) = w \cdot v_i(\pi) + (1-w) \cdot \max_{f \in F_i \cup \{i\}}(\{v_f(\pi)\})$ and $u_i^{fa}(\pi) = w \cdot v_i(\pi) + (1-w) \cdot \frac{1}{|F_i|+1}\sum_{f \in F_i \cup \{i\}} v_f(\pi)$ respectively, where $w \in (0,1)$ is selfishness degree parameter.

Given two coalitions $\pi_1$ and $\pi_2$, $\pi_1 \succ_i \pi_2$ occurs if and only if $u_i(\pi_1) \leq u_i(\pi_2)$. Donahue et al. [15] have established the sufficient conditions for achieving *core-stability* under the assumption of selfish clients, as in Lemma 2. Building upon their findings, we further introduce these conditions to cover scenarios of altruistic behaviors, as in Corollary 1.

**Lemma 2** *(Lemma 10 in [15]) When clients are purely selfish, for a set of clients with $n_i \leq \frac{\mu_e}{\sigma^2}$, the grand coalition $\pi_g$ is core-stable.*

**Corollary 1** *For a set of players where $n_i \leq \frac{\mu_e}{\sigma^2}$, the grand coalition $\pi_g$ remains core-stable when the clients are either purely altruistic or friendly altruistic. The proof is provided in Appendix A.2.*

### 4.2 Altruism hedonic game vs. altruism coalition formation game

Existing research [12, 14, 32] views FL as a hedonic game, where the players only care about the identity of the players in their coalition. Correspondingly, within the framework of the altruism hedonic game (AHG), the set of friends influencing the $i$-th client's participation decision is confined to $F_i \rightarrow F_i \cap \pi_i$, where $\pi_i$ represents the coalition to which the $i$-th client belongs. In AHG, excluding some friends from a client's altruistic behavior can result in a core-stable coalition structure that is non-Pareto optimal in terms of local errors. The non-Pareto optimal indicates that an alternative coalition structure can be established where each client encounters an error not higher than that observed within the core-stable coalition structure, with at least one client experiencing a lower error. Such a core-stable coalition structure contradicts the assumption of rationality among clients. For example, Table 1 shows the error and utility of 4 friendly welfare altruistic ($w = 0.5$) clients in the mean estimation task. In a fully connected friends-relationship network (Relation I in Figure 1), we first employ the AHG model to analyze the FL system. We recognize that the core-stable coalition structure $\Pi_{\text{stable}} = \{\{1,2\},\{3\},\{4\}\}$ is non-Pareto optimal in local errors as there exists another alternative structure, i.e., $\Pi_{alter} = \{\{1,2,3\},\{4\}\}$, in which the errors are not greater than in $\Pi_{stable}$ (i.e., $err_i(\Pi_{alter}) \leq err_i(\Pi_{stable}), \forall i \in \{1,2,3,4\}$) and at least one client exhibits less error than that in $\Pi_{stable}$ (i.e., $err_i(\Pi_{alter}) \leq err_i(\Pi_{stable}), \forall i \in \{1,2,3\}$). To ensure a core-stable coalition structure that guarantees Pareto optimality and adheres to the rationality assumptions of the players, we model the FL process with altruistic behavior as an altruism coalition formation game (ACFG), where a client's utility

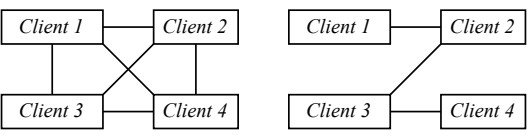

Figure 1: Friends-relationship networks: fully connected relation I (left) and partially connected relation II (right).

is derived from the entire friend-network, regardless of whether the friends belong to the same coalition as the client. The ACFG fully captures the impact of altruistic behavior on a client's decision to stay or leave a coalition; for example, a client may opt to leave the current coalition to satisfy the needs of friends in other coalitions, a scenario that AHG can not capture. As shown in Table 1, the core-stable coalition structure of a fully connected friends-relationship network under ACFG, $\Pi_{\text{stable}} = \{\{1,2,3,4\}\}$, is Pareto-optimal in local errors across clients.

**Proposition 1** *(Pareto-optimality in error, Appendix A.3) Consider the FL system described as an ACFG, a core-stable coalition structure is also Pareto-optimal in local errors across all clients.*

Table 1: The error and utility for friendly welfare altruistic clients within fully connected and partially connected friendship networks under AHG and ACFG frameworks, respectively.

| Coalition Structure | Error (=$u^{ps}$) | | | | Utility $u^{fa}$ in AHG (Relation I) | | | | Utility $u^{fa}$ in ACFG (Relation I) | | | | Utility $u^{fa}$ in ACFG (Relation II) | | | |
|---|---|---|---|---|---|---|---|---|---|---|---|---|---|---|---|---|
| | $err_1$ | $err_2$ | $err_3$ | $err_4$ | $u_1$ | $u_2$ | $u_3$ | $u_4$ | $u_1$ | $u_2$ | $u_3$ | $u_4$ | $u_1$ | $u_2$ | $u_3$ | $u_4$ |
| {1} | 2.0 | / | / | / | 2.0 | / | / | / | 2.0 | / | / | / | 2.0 | / | / | / |
| {2} | / | 2.0 | / | / | / | 2.0 | / | / | / | 2.0 | / | / | / | 2.0 | / | / |
| {3} | / | / | 1.0 | / | / | / | 1.0 | / | / | / | 1.22 | / | / | / | 1.22 | / |
| {4} | / | / | / | 0.666 | / | / | / | 0.666 | / | / | / | 1.020 | / | / | / | 0.770 |
| {1,2} | 1.5 | 1.5 | / | / | 1.5 | 1.5 | / | / | 1.5 | 1.5 | / | / | 1.5 | 1.5 | / | / |
| {2,3} | / | 1.555 | 0.888 | / | / | 1.555 | 1.222 | / | / | 1.590 | 1.256 | / | / | 1.590 | 1.222 | / |
| {3,4} | / | / | 1.12 | 0.72 | / | / | 1.12 | 0.92 | / | / | 1.31 | 1.11 | / | / | 1.31 | 0.92 |
| {1,3} | 1.555 | / | 0.888 | / | 1.555 | / | 1.222 | / | 1.590 | / | 1.256 | / | 1.590 | / | 1.256 | / |
| {1,4} | 1.625 | / | / | 0.625 | 1.625 | / | / | 1.125 | 1.625 | / | / | 1.125 | 1.625 | / | / | 0.756 |
| {2,4} | / | 1.625 | / | 0.625 | / | 1.625 | / | 1.125 | / | 1.625 | / | 1.125 | / | 1.625 | / | 0.756 |
| {1,2,3} | 1.375 | 1.375 | 0.875 | / | 1.375 | 1.375 | 1.125 | / | 1.375 | 1.375 | 1.125 | / | 1.375 | 1.375 | 1.125 | / |
| {1,2,4} | 1.44 | 1.44 | / | 0.64 | 1.44 | 1.44 | / | 1.04 | 1.44 | 1.44 | / | 1.04 | 1.44 | 1.44 | / | 0.82 |
| {1,3,4} | 1.388 | / | 1.055 | 0.722 | 1.388 | / | 1.222 | 1.055 | 1.694 | / | 1.527 | 1.361 | 1.694 | / | 1.527 | 0.888 |
| {2,3,4} | / | 1.388 | 1.055 | 0.722 | / | 1.388 | 1.222 | 1.055 | / | 1.694 | 1.527 | 1.361 | / | 1.694 | 1.222 | 0.888 |
| {1,2,3,4} | 1.306 | 1.306 | 1.020 | 0.734 | 1.306 | 1.306 | 1.163 | 1.020 | 1.306 | 1.306 | 1.163 | 1.020 | 1.306 | 1.306 | 1.163 | 0.877 |

# 5 Egalitarian fairness bound in core-stable federated learning

## 5.1 Does egalitarian fairness lead to instability?

To achieve egalitarian fairness in FL [4, 5, 20, 6, 7, 8], resource-advantaged clients may sacrifice local error to improve error uniformity of global model across all participants, creating a motivation for these resource-advantaged clients to leave the FL framework. In this part, we initially reveal the factors that impact the emergence of issue "*egalitarian fairness leads to instability*". As demonstrated in Table 1, the most egalitarian fair coalition structure is grand coalition, $\Pi_G = \{\pi_g\} = \{\{1, 2, 3, 4\}\}$, with a fairness bound $\lambda = 1.306/0.734 \approx 1.78$.

When all clients are purely selfish and focus solely on minimizing their local error, the coalition structure $\Pi_G$ is not core-stable. The current core stable coalition structure is $\Pi_{stable} = \{\{1, 2, 3\}, \{4\}\}$ with a fairness bound $\lambda = 1.375/0.666 \approx 2.06$. However, when clients exhibit altruistic behavior, and the friends-relationship network is fully connected (Relation I), the most egalitarian fair coalition structure $\Pi_{stable} = \Pi_G = \{\{1, 2, 3, 4\}\}$ with a fairness bound $\lambda = 1.306/0.734 \approx 1.78$ is also core stable. The finding demonstrates whether "*egalitarian fairness leads to instability*" is influenced by the clients' behavior. Furthermore, when the friends-relationship network among the clients changes to partially connected (Relation II), leading to the coalition structure $\Pi_G = \{\{1, 2, 3, 4\}\}$ becoming not core-stable anymore and the current core stable coalition

Table 2: Notation Definitions.

| Notation | Description |
|---|---|
| $\pi_c$ | The complement coalition of a coalition $\pi_s$: $\pi_c = \pi_g \setminus \pi_s$. |
| $N_s$ | The sum of the dataset sizes in $\pi_s$: $N_s = \sum_{i \in \pi_s} n_i$. |
| $N_c$ | The sum of the dataset sizes in $\pi_c$: $N_c = \sum_{i \in \pi_c} n_i$. |
| $N_g$ | The sum of the dataset sizes in the grand coalition: $N_g = \sum_{i \in \pi_g} n_i$. |
| $m$ | The index of the client with the smallest dataset size in $\pi_g$: $m = \arg\min_{i \in \pi_g}\{n_i\}$. |
| $l$ | The index of the client with the largest dataset size in $\pi_g$: $l = \arg\max_{i \in \pi_g}\{n_i\}$. |

structure is $\Pi_{stable} = \{\{1, 2, 3\}, \{4\}\}$ with a fairness bound $\lambda = 1.375/0.666 \approx 2.06$. The observation demonstrates that, under altruistic behaviors, the diverse friends-relationship networks also impact whether "*egalitarian fairness leads to instability*".

From the above analysis, we observe a varying optimal fairness achieved within the core-stable coalition under diverse client behaviors and friends-relationship networks, highlighting a significant problem to guide the setting of fairness in FL: *for an FL system, what is the optimal egalitarian fairness that can be achieved without compromising the core-stability of the FL system?*

## 5.2 Optimal egalitarian fairness bound

We define a distance function to measure the dataset size of a client relative to all other clients within the same coalition $\pi$,

$$d(\pi, n_j) = \left(\sum_{i \in \pi} n_i^2 - n_j^2\right) + \left(\sum_{i \in \pi} n_i - n_j\right)^2. \tag{3}$$

Given the grand coalition $\pi_g$ in an FL system consists of a set of $N$ clients with local dataset size $n_i \leq \frac{\mu_e}{\sigma^2}$ and based on the properties of mean estimation task in Corollary 1, the grand coalition $\pi_g$ is core-stable. In this context, we give the tight egalitarian fairness bound achieved by $\pi_g$ in Proposition $2\sim 6$. For any coalition $\pi_s \subset \pi_g$, we unify the common notations to represent the data size within this subset and its complement, as well as the specific individual indices in Table 2.

**Proposition 2** *(Optimal egalitarian fairness under purely selfish behaviors, Appendix A.4.1) Considering all clients act purely selfish, the grand coalition $\pi_g$ remains core-stable if the achieved egalitarian fairness is bounded by:*

$$\lambda \geq \max_{\pi_s \subset \pi_g} \left\{ \frac{N_s{}^2}{N_g{}^2} \cdot \frac{N_g \cdot n_l + d(\pi_g, n_m)}{N_s \cdot n_l + d(\pi_s, n_{k_{\pi_s}})} \right\}, \ where \ k_{\pi_s} = \arg\min_{i \in \pi_s} \{n_i\}. \tag{4}$$

Proposition 2 demonstrates that an increase in the heterogeneity of clients' local dataset sizes—reflected by *the difference between the smallest dataset size overall and those within any given subset coalition*—the achievable egalitarian fairness of a core-stable grand coalition becomes poorer. Furthermore, we can obtain a sufficient condition for achieving strict egalitarian fairness ($\lambda = 1$) in a core-stable coalition with purely selfish clients based on Equation (4).

**Corollary 2** *The core-stable grand coalition $\pi_g$, comprising all selfish clients, can asymptotically achieve strict egalitarian fairness, provided that the local dataset sizes of all clients are equal. The proof is given in Appendix A.4.2.*

**Proposition 3** *(Optimal egalitarian fairness under purely welfare altruistic behaviors, Appendix A.4.3) Considering all clients act purely welfare altruistic, the grand coalition $\pi_g$ remains core-stable if the achieved egalitarian fairness is bounded by:*

$$\lambda \geq \max_{\pi_s \in \pi_g} \left\{ \min\left( \frac{N_s{}^2}{N_g{}^2} \cdot \frac{N_g \cdot n_l + d(\pi_g, n_m)}{N_s \cdot n_l + d(\pi_s, f_{\pi_s,1}^{opt})}, \frac{N_c{}^2}{N_g{}^2} \cdot \frac{N_g \cdot n_l + d(\pi_g, n_m)}{N_c \cdot n_l + d(\pi_c, f_{\pi_s,2}^{opt})} \right) \right\},$$

$$where$$

$$k_{\pi_s,1} = \arg\min_{i \in \pi_s} \left\{ \min_{f \in F_i \cap \pi_s} n_f \right\}, k_{\pi_s,2} = \arg\min_{i \in \pi_s} \left\{ \min_{f \in F_i \cap \pi_c} n_f \right\},$$

$$f_{\pi_s,1}^{opt} = \arg\min_{f \in F_{k_{\pi_s,1}} \cap \pi_s} n_f, f_{\pi_s,2}^{opt} = \arg\min_{f \in F_{k_{\pi_s,2}} \cap \pi_c} n_f. \tag{5}$$

Proposition 3 demonstrates that the achieved egalitarian fairness declines as the gap between *the smallest dataset size overall and the smallest dataset size within any given friends-relationship network* increases. This gap is significantly smaller than that in purely selfish cases, thereby facilitating improved egalitarian fairness and creating a more relaxed condition for achieving strict egalitarian fairness as in Corollary 3.

**Corollary 3** *The core-stable grand coalition $\pi_g$, consisting of purely welfare clients, can asymptotically achieve strict egalitarian fairness if all clients are friends with the client possessing the smallest dataset size and $N_g \to \infty$. The proof is given in Appendix A.4.4.*

**Proposition 4** *(Optimal egalitarian fairness under purely equal altruistic behaviors, Appendix A.4.5) Considering all clients act purely equal altruistic, the grand coalition $\pi_g$ remains core-stable if the achieved egalitarian fairness is bounded by:*

$$\lambda \geq \max_{\pi_s \in \pi_g} \left( \frac{|F_{k_{\pi_s}}| \cdot N_s{}^2 N_c{}^2}{N_g{}^2} \cdot \frac{N_g \cdot n_l + d(\pi_g, n_m)}{\mathbf{Q}} \right),$$

$$where$$

$$k_{\pi_s} = \arg\min_{i \in \pi_s} \frac{1}{|F_i|} \left( \sum_{f \in F_i \cap \pi_s} n_f + \sum_{f \in F_i \cap \pi_c} n_f \right),$$

$$\mathbf{Q} = N_c^2 \cdot \sum_{f \in F_{k_{\pi_s}} \cap \pi_s} (N_s \cdot n_l + d(\pi_s, n_f)) + N_s^2 \cdot \sum_{f \in F_{k_{\pi_s}} \cap \pi_c} (N_c \cdot n_l + d(\pi_c, n_f)). \tag{6}$$

Proposition 4 shows that the egalitarian fairness bound for purely equal altruistic clients is influenced by the gap between *the smallest dataset size overall and the weighted sum of dataset sizes within any given friends-relationship network*. According to Propositions 5 and 6, the egalitarian fairness bounds in the context of friendly altruism behavior are shaped by two factors: (1) the heterogeneity

of clients' local dataset sizes and (2) the difference between the smallest dataset size in the grand coalition and either the smallest dataset size or weighted sum of dataset sizes within established friends-relationship networks. The selfishness degree parameter ($w$) balances the relative significance of these two factors.

**Proposition 5** *(Optimal egalitarian fairness under friendly welfare altruistic behaviors, Appendix A.4.6) Considering all clients act friendly welfare altruistic, the grand coalition $\pi_g$ remains core-stable if the achieved egalitarian fairness is bounded by:*

$$\lambda \geq \max_{\pi_s \in \pi_g} \left\{ \min \left( \frac{N_s^2}{N_g^2} \cdot \frac{N_g \cdot n_l + d(\pi_g, n_m)}{\mathbf{Q}_1}, \frac{N_s^2 N_c^2}{N_g^2} \cdot \frac{N_g \cdot n_l + d(\pi_g, n_m)}{\mathbf{Q}_2} \right) \right\},$$

*where*

$$
\begin{aligned}
k_{\pi_s,1} &= \arg\min_{i \in \pi_s} \left\{ w \cdot n_i + (1-w) \cdot \min_{f \in F_i \cap \pi_s \cup \{i\}} n_f \right\}, \\
k_{\pi_s,2} &= \arg\min_{i \in \pi_s} \left\{ w \cdot n_i + (1-w) \cdot \min_{f \in F_i \cap \pi_c} n_f \right\}, \\
f_{\pi_s,1}^{opt} &= \arg\min_{f \in F_{k_{\pi_s,1}} \cap \pi_s \cup \{k_{\pi_s,1}\}} n_f, \quad f_{\pi_s,2}^{opt} = \arg\min_{f \in F_{k_{\pi_s,2}} \cap \pi_c} n_f, \\
\mathbf{Q}_1 &= N_s \cdot n_l + w \cdot d(\pi_s, n_{k_{\pi_s,1}}) + (1-w) \cdot d(\pi_s, f_{\pi_s,1}^{opt}), \\
\mathbf{Q}_2 &= N_c^2 \cdot w \cdot \left( N_s \cdot n_l + d(\pi_s, n_{k_{\pi_s,2}}) \right) + N_s^2 \cdot (1-w) \cdot \left( N_c \cdot n_l + d(\pi_c, f_{\pi_s,2}^{opt}) \right).
\end{aligned}
\tag{7}
$$

**Proposition 6** *(Optimal egalitarian fairness under friendly equal altruistic behaviors, Appendix A.4.7) Considering all clients act friendly equal altruistic, the grand coalition $\pi_g$ remains core-stable if the achieved egalitarian fairness is bounded by:*

$$\lambda \geq \max_{\pi_s \in \pi_g} \left( \frac{(|F_{k_{\pi_s}}|+1) \cdot N_s^2 \cdot N_c^2}{N_g^2} \cdot \frac{N_g \cdot n_l + d(\pi_g, n_m)}{\mathbf{Q}} \right),$$

*where*

$$
\begin{aligned}
k_{\pi_s} &= \arg\min_{i \in \pi_s} \left( w \cdot n_i + (1-w) \cdot \frac{1}{|F_i|+1} \cdot \left( \sum_{f \in F_i \cap \pi_s \cup \{i\}} n_f + \sum_{f \in F_i \cap \pi_c} n_f \right) \right), \\
\hat{F}_s &= F_{k_{\pi_s}} \cap \pi_s \cup \{k_{\pi_s}\}, \quad \hat{F}_c = F_{k_{\pi_s}} \cap \pi_c, \\
\mathbf{Q} &= w \cdot (|F_{k_{\pi_s}}|+1) \cdot N_c^2 \cdot (N_s \cdot n_l + d(\pi_s, n_{k_{\pi_s}})) +
\end{aligned}
\tag{8}
$$

$$(1-w) \cdot \left( N_c^2 \cdot \sum_{f \in \hat{F}_s} (N_s \cdot n_l + d(\pi_s, n_f)) + N_s^2 \cdot \sum_{f \in \hat{F}_c} (N_c \cdot n_l + d(\pi_c, n_f)) \right).$$

The given propositions establish tight egalitarian fairness bounds in core-stable FLs and demonstrate that the achievable egalitarian fairness for a collection of clients varies depending on the clients' behaviors and friends-relationship networks. Furthermore, these propositions can be adapted to situations where clients exhibit heterogeneous behaviors within an FL system as follows.

### 5.3 Heterogeneous behaviors

When client behaviors exhibit heterogeneity in an FL system, the achievable egalitarian fairness for a core-stable grand coalition $\pi_g$ is determined by the maximum egalitarian fairness bounds calculated across sub-coalitions formed by clients with homogeneous behaviors.

**Example 1** *An example to calculate the achievable egalitarian fairness bound under heterogeneous behaviors is as follows: for a set of $N$ clients, where clients $i \in \mathbf{C} = \{1, 2, ..., S\}$ act selfishly and the remaining act purely welfare altruistic, the achieved egalitarian fairness of $\pi_g$ is bounded by,*

$$\lambda \geq \max_{\pi_s \subset \pi_g} \left\{ \max \left\{ \frac{N_s^2}{N_g^2} \cdot \frac{N_g \cdot n_l + d(\pi_g, n_m)}{N_s \cdot n_l + d(\pi_s, n_{k_{selfish}})}, \min \left( \begin{matrix} \frac{N_s^2}{N_g^2} \cdot \frac{N_g \cdot n_l + d(\pi_g, n_m)}{N_s \cdot n_l + d(\pi_s, f_{altruistic,1}^{opt})}, \\ \frac{N_c^2}{N_g^2} \cdot \frac{N_g \cdot n_l + d(\pi_g, n_m)}{N_c \cdot n_l + d(\pi_c, f_{altruistic,2}^{opt})} \end{matrix} \right) \right\} \right\},$$

*where*

$$
\begin{aligned}
k_{selfish} &= \arg\min_{i \in \pi_s \cap \mathbf{C}} \{n_i\}, \\
k_{altruistic,1} = \arg\min_{i \in \pi_s \setminus \mathbf{C}} \left\{ \min_{f \in F_i \cap \pi_s} n_f \right\}, & \quad k_{altruistic,2} = \arg\min_{i \in \pi_s \setminus \mathbf{C}} \left\{ \min_{f \in F_i \cap \pi_c} n_f \right\}, \\
f_{altruistic,1}^{opt} = \arg\min_{f \in F_{k_{altruistic,1}} \cap \pi_s} n_f, & \quad f_{altruistic,2}^{opt} = \arg\min_{f \in F_{k_{altruistic,2}} \cap \pi_c} n_f.
\end{aligned}
\tag{9}
$$

## 6 Evaluation

In following experiments, we validate the tightness and applicability of the analyzed fairness bounds for the given tasks.

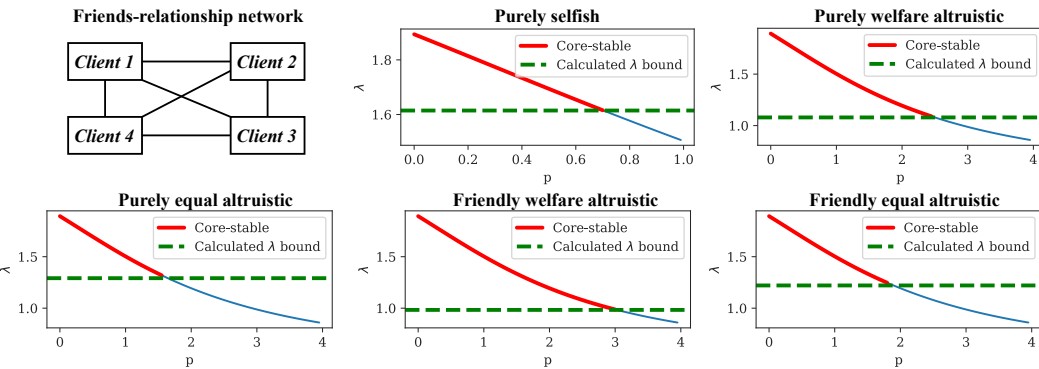

Figure 2: Fully connected: theoretically derived egalitarian fairness bounds (green dashed line) align with empirically achieved egalitarian fairness within the core-stable grand coalition (red solid line) under different client behaviors.

## 6.1 Settings

To conduct the experiments, we first give a fair FL framework to implement egalitarian fairness, where a coefficient $p$ is introduced to increase the aggregation weight of clients who received higher local errors: $\theta_C = \frac{\sum_{i \in C} \left( n_i \cdot err_i^p \cdot \theta_i \right)}{\sum_{i \in C} \left( n_i \cdot err_i^p \right)}$. The preceding discussions are independent of the fair FL framework chosen. We follow up on the task models proposed by Donahue et al. [12]. The first case is a mean estimation task involving a fixed set of 4 players. Each player has a fixed number of samples, specifically $\{20, 40, 50, 100\}$. Each player draws their local data samples from their true distribution with parameters $\mathcal{Y}_i \sim (\theta_i, \epsilon_i)$, where the samples are independent and identically distributed (i.i.d.). Here, $\theta_i \sim \mathcal{N}(\mu = 0, \sigma^2 = 1)$ and $\epsilon_i \sim 125 \times Beta(a = 8, b = 2)$. To further validate the adaptability of this theoretical results, we introduce additional linear regression task with a fixed set of 3 players. Each player has a fixed number of samples, specifically $\{50, 100, 200\}$. Each player draws 2-dimensional input features from their own input distribution, $\mathcal{X}_i^1, \mathcal{X}_i^2 \sim \mathcal{N}(\mu = 0, \sigma^2 = 1)$. The output label is drawn from $\mathcal{Y}_i \sim 250 \times Beta(a = 8, b = 2)$. All experiments are conducted on Intel(R) Xeon(R) Gold 5318Y CPU @ 2.10GHz.

## 6.2 Results

We conduct the effectiveness and tightness validation of the theoretical egalitarian fairness bound across diverse scenarios include (1) different client behaviors (Figure 2), (2) different structures of friends-relationship networks (Figure 3). The results demonstrate the alignment between our theoretically derived egalitarian fairness bound (green dashed line) and the empirically achieved egalitarian fairness within the core-stable grand coalition (red solid line) under both fully connected and partially connected friends-relationship networks.

In Appendix A.5, we conduct additional experiments to further validate the effectiveness of the theoretical results under (3) heterogeneous client behaviors (Figure 4) and (4) additional regression task (Figure 5).

## 7 Discussion and Limitations

In this section, we outline the scope of our work and identify several limitations of our theoretical and experimental results, which could serve as potential starting points for future work.

**More complex task scenarios.** This work paves the way for future research within the domain of FL. Initially, it would be insightful to explore the impact of other notions of fairness, such as proportional fairness [33]—where a client's payoff is proportional to its data contribution—on the stability of FL. Moreover, for theoretical proofs, our work is built upon the exact-error task models developed by Donahue et al. [12]; future work could be extended to more complex collaborative training tasks. Initially, we chose the maximum and linear weighting of errors to construct the utility functions

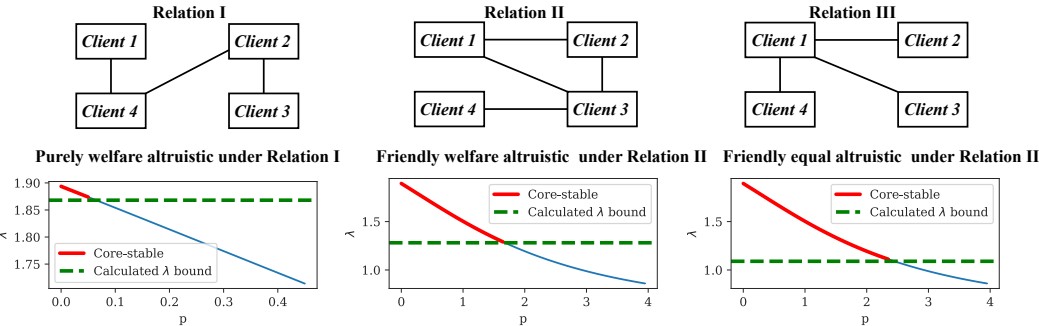

Figure 3: Partially connected: theoretically derived egalitarian fairness bounds (green dashed line) align with empirically achieved egalitarian fairness within the core-stable grand coalition (red solid line) under different client behaviors.

because they are simple and typical examples for theoretical analysis. Moving forward, our key theoretical ideas can be extended to a broader class of utility functions in the form of generalized mean, which encompasses many prominent social welfare functions, i.e., weighted power-mean welfare function [34] (as in Appendix A.4.8).

**Incentive Mechanisms and Client Behavior Dynamics.** As we discover the bounds of egalitarian fairness that a stable coalition can achieve, a practical question arises: How can we correspondingly design suitable incentive mechanisms to retain clients in the coalition when high egalitarian fairness is a mandatory requirement? Given that altruism is closely related to stability when pursuing fairness, further exploration into how various human behaviors—such as reciprocity, bounded rationality, risk aversion, and risk tolerance—impact the stability of FLs in achieving specific objectives would be of profound interest.

**Overfitting.** Centering on exploring the relationship between egalitarian fairness and stability, we use the stylized model in Lemma 1, which provides a closed-form error to derive precise relations and generate insights into fairness settings in fair FLs. However, overfitting [35] in machine learning leads to uncertainty in the error outlined in Equation 1, which is influenced by model structure and parameters, choice of training algorithms, and the specific data distributions of clients, etc. Extending our theoretical analysis to establish fairness bounds in specific model states, such as overfitting, is considerably more complex and remains an open area of research; however, our approach may still provide a valuable foundation for further exploration.

# 8 Conclusion

In this work, we have rigorously answered a previously unexplored but critical question: *Does egalitarian fairness lead to instability?* Through our analysis, we have explored the influence of clients' altruistic behaviors and the configuration of the friend-relationship network on the achievable egalitarian fairness within a core-stable federated learning (FL) coalition. Our research has identified the optimal egalitarian fairness bounds without compromising core stability from a theoretical standpoint. The outcomes of our research can be leveraged to establish appropriate egalitarian fairness in FL implementation, which plays a crucial role in improving the alignment of FL processes with societal welfare and ethical standards.

# 9 Acknowledgements

This work was supported by Key Programs of Guangdong Province under Grant 2021QN02X166. Any opinions, findings, and conclusions or recommendations expressed in this material are those of the author(s) and do not necessarily reflect the views of the funding parties.

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

# A  Appendix

## A.1  Monotonicity of error with dataset size

**Lemma 3** *(Monotonicity) For any two clients within the same coalition, $i, j \in \pi$, if $n_i > n_j$, $err_i(\pi) < err_j(\pi)$. Conversely, if $err_i < err_j$, $n_i > n_j$.*

**Proof** *Based on the exact errors in Lemma 1, it's natural that if $n_i > n_j$, $err_i(\pi) < err_j(\pi)$. Denote $N_s = \sum_{k \in \pi} n_k$, $\hat{N}_s = \sum_{k \in \pi} n_k^2$, if $err_i < err_j$, it holds that,*

$$\frac{\mu_e}{N_s} + \sigma^2 \cdot \frac{\hat{N}_s - n_i^2 + (N_s - n_i)^2}{N_s^2} < \frac{\mu_e}{N_s} + \sigma^2 \cdot \frac{\hat{N}_s - n_j^2 + (N_s - n_j)^2}{N_s^2}. \quad (10)$$
$$\rightarrow \frac{-2\sigma^2 \cdot N_s \cdot n_i}{N_s^2} < \frac{-2\sigma^2 \cdot N_s \cdot n_j}{N_s^2} \rightarrow n_i > n_j$$

## A.2  Core-stable conditions

**Corollary 1** For a set of players where $n_i \leq \frac{\mu_e}{\sigma^2}$, the grand coalition $\pi_g$ remains core-stable when the clients are either purely altruistic or friendly altruistic.

**Proof** *In a grand coalition $\pi_g$ comprising $N$ clients, each client holds a local dataset with size $n_i \leq \frac{\mu_e}{\sigma^2}$. Based on Lemma 2, when $n_i \leq \frac{\mu_e}{\sigma^2}$, it holds that $err_i(\pi_g) \leq err_i(\pi_s)$. Whether within $\pi_g$ or a subset $\pi_s \subset \pi_g$, the $i$-th purely welfare altruistic client focuses on the same friend, that is, the $k$-th client, where $k = \arg\min_{f \in F_i} \{n_f\}$.*

$$err_k(\pi_g) \leq err_k(\pi_s) \Rightarrow \min_{f \in F_i} \left(\{err_f(\pi_g)\}\right) \leq \min_{f \in F_i} \left(\{err_f(\pi_s)\}\right), \forall i, \pi_s \subset \pi_g. \quad (11)$$

*Therefore, the $\pi_g$ is core-stable for purely welfare altruistic clients. For the purely equal altruistic client, the optimal individual error under $\pi_g$ also leads to the optimality of the average multi-individual error and the following equation holds:*

$$\frac{1}{|F_i|} \sum_{f \in F_i} err_f(\pi_g) \leq \frac{1}{|F_i|} \sum_{f \in F_i} err_f(\pi_s), \forall i, \pi_s \subset \pi_g. \quad (12)$$

*Therefore, the $\pi_g$ is core-stable for purely equal altruistic clients.*

*The utilities of friendly altruistic clients are a weighted combination of the utilities under selfish and purely altruistic behaviors. According to the established core-stability when clients are selfish and purely altruism, the condition $n_i \leq \frac{\mu_e}{\sigma^2}$ facilitates utility minimization within the grand coalition $\pi_g$ for clients with selfish and purely altruistic behaviors. Thus, this condition also leads to the utilities within the grand coalition $\pi_g$ being minimized for clients exhibiting friendly altruistic behaviors.*

## A.3  Pareto-Optimality

**Proportion 1** (Pareto-Optimality in error) Consider the FL system described as an ACFG, a core-stable coalition structure is also Pareto-optimal in local errors across all clients.

**Proof** *We prove the Proportion 1 by contradiction. Consider $N$ clients. Assume that a core-stable coalition structure $\Pi_{CS}$ is not Pareto-optimal. Then, there exists a Pareto-optimal coalition structure $\Pi_{PO}$ that satisfies the following conditions:*

$$err_i \left(\Pi_{CS}\right) \geq err_i \left(\Pi_{PO}\right), \forall i \in \{N\}, \\ err_j \left(\Pi_{CS}\right) > err_j \left(\Pi_{PO}\right), \exists j \in \{N\}. \quad (13)$$

*For clients exhibit altruism with selfishness degree parameter $w \in [0, 1]$, it holds that:*

$$w \cdot v_j \left(\Pi_{CS}\right) + (1 - w) \cdot v_{k \in F_j \cup \{j\}} \left(\Pi_{CS}\right) > w \cdot v_j \left(\Pi_{PO}\right) + (1 - w) \cdot v_{k \in F_j \cup \{j\}} \left(\Pi_{PO}\right), \quad (14)$$

*which is contradicts the assumption of $\Pi_{CS}$ is core-stable. Based on the above, a core-stable federation structure is also Pareto-optimal under ACFG.*

### A.4 Egalitarian fairness bound

#### A.4.1 Proof of Proposition 2

**Proposition 2** (Optimal egalitarian fairness under purely selfish behaviors) Considering all clients act purely selfish, the grand coalition $\pi_g$ remains core-stable if the achieved egalitarian fairness is bounded by:

$$\lambda \geq \max_{\pi_s \subset \pi_g} \left\{ \frac{N_s{}^2}{N_g{}^2} \cdot \frac{N_g \cdot n_l + d(\pi_g, n_m)}{N_s \cdot n_l + d(\pi_s, n_{k_{\pi_s}})} \right\},$$

*where*

$$k_{\pi_s} = \arg\min_{i \in \pi_s} \{n_i\}.$$

(15)

**Proof** *In a coalition structure with $N$ clients, denoted by $\pi_g = \{1, 2, ..., N\}$, where each client possesses a local dataset of size $n_j$, the concept of $\lambda$-egalitarian fairness stipulates that $\frac{err_i(\pi_g)}{err_j(\pi_g)} \geq \lambda$, $n_i \leq n_j$. In a context where clients are purely selfish, the coalition $\pi_g$ maintains core-stable if, for any potential sub-coalition $\pi_s \subset \pi_g$, there is at least one client who prefers $\pi_g$ over $\pi_s$, this means that $\exists i \in \pi_s, u_i(\pi_g) \leq u_i(\pi_s)$, or equivalently, $\exists i \in \pi_s, err_i(\pi_g) \leq err_i(\pi_s)$. According to the egalitarian fairness definition, it holds that,*

$$err_i(\pi_g) \leq \frac{\max\{err_j(\pi_g)\}_{j=1}^N}{\lambda}.$$

(16)

*To satisfy for $\forall \pi_s \subset \pi_g, \exists i \in \pi_s, err_i(\pi_g) \leq err_i(\pi_s)$, the lowest $\lambda$ is bounded by Equation (17).*

$$\frac{\max\{err_i(\pi_g)\}_{i=1}^N}{\lambda} \leq \min_{\pi_s \subset \pi_g} \left\{ \max_{i \in \pi_s}\{err_i(\pi_s)\} \right\}.$$

(17)

*Based on Lemma 3, we denote $m = \arg\max_{i \in \pi_g}\{err_i(\pi_g)\} = \arg\min_{i \in \pi_g}\{n_i\}$ and $k_{\pi_s} = \arg\max_{i \in \pi_s}\{err_i\} = \arg\min_{i \in \pi_s}\{n_i\}$, then the above equation is equivalent to,*

$$\frac{err_m(\pi_g)}{\lambda} \leq \min_{\pi_s \subset \pi_g} \left\{ err_{k_{\pi_s}}(\pi_s) \right\}.$$

(18)

*We first give the lowest fairness bound $\lambda$ for a specific $\pi_s$,*

$$\frac{err_m(\pi_g)}{\lambda} \leq err_{k_{\pi_s}}(\pi_s).$$

(19)

*Based on Lemma 1, the above equation can be transformed as,*

$$\frac{\mu_e}{\sum_{i \in \pi_g} n_i} + \sigma^2 \cdot \frac{\sum_{i \in \pi_g, i \neq m} n_i^2 + \left(\sum_{i \in \pi_g, i \neq m} n_i\right)^2}{\left(\sum_{i \in \pi_g} n_i\right)^2}$$
$$\leq \lambda \cdot \left( \frac{\mu_e}{\sum_{i \in \pi_s} n_i} + \sigma^2 \cdot \frac{\sum_{i \in \pi_s, i \neq k_{\pi_s}} n_i^2 + \left(\sum_{i \in \pi_s, i \neq k_{\pi_s}} n_i\right)^2}{\left(\sum_{i \in \pi_s} n_i\right)^2} \right).$$

(20)

*Denote $N_s = \sum_{i \in \pi_s} n_i$ and $N_g = \sum_{i \in \pi_g} n_i$, we have,*

$$\frac{\mu_e}{N_g} + \sigma^2 \cdot \frac{\sum_{i \in \pi_g, i \neq m} n_i^2 + (N_g - n_m)^2}{N_g^2} \leq \lambda \cdot \left( \frac{\mu_e}{N_s} + \sigma^2 \cdot \frac{\sum_{i \in \pi_s, i \neq k_{\pi_s}} n_i^2 + (N_s - n_{k_{\pi_s}})^2}{N_s^2} \right).$$

(21)

*Multiplying each side of the above equation by $N_g^2 N_s^2$,*

$$\mu_e \cdot N_g \cdot N_s{}^2 + \sigma^2 \cdot N_s{}^2 \cdot \left( \sum_{i \in \pi_g, i \neq m} n_i^2 + (N_g - n_m)^2 \right)$$
$$\leq \lambda \cdot \left( \mu_e \cdot N_s \cdot N_g{}^2 + \sigma^2 \cdot N_g{}^2 \cdot \left( \sum_{i \in \pi_s, i \neq k_{\pi_s}} n_i^2 + \left(N_s - n_{k_{\pi_s}}\right)^2 \right) \right).$$

(22)

*Thus, the lowest fairness bound $\lambda$ satisfies,*

$$\lambda \geq \frac{\mu_e \cdot N_g \cdot N_s{}^2 + \sigma^2 \cdot N_s{}^2 \cdot \left( \sum_{i \in \pi_g, i \neq m} n_i^2 + (N_g - n_m)^2 \right)}{\mu_e \cdot N_s \cdot N_g{}^2 + \sigma^2 \cdot N_g{}^2 \cdot \left( \sum_{i \in \pi_s, i \neq k_{\pi_s}} n_i^2 + \left(N_s - n_{k_{\pi_s}}\right)^2 \right)} \to f_{RHS}.$$

(23)

Taking the derivative of $f_{RHS}$ with respect to $\sigma$, we have:

$$
\begin{aligned}
f'_{RHS,\sigma} &= \frac{\mu_e \cdot N_g \cdot N_s{}^2 + \sigma^2 \cdot N_s{}^2 \cdot \left( \sum_{i \in \pi_g, i \neq m} n_i^2 + (N_g - n_m)^2 \right)}{\left( \mu_e \cdot N_s \cdot N_g{}^2 + \sigma^2 \cdot N_g{}^2 \cdot \left( \sum_{i \in \pi_s, i \neq k_{\pi_s}} n_i^2 + \left( N_s - n_{k_{\pi_s}} \right)^2 \right) \right)^2} \\
&= \frac{2\sigma \cdot \mu_e \cdot N_s^2 \cdot N_g^2 \cdot \left( N_s \cdot \left( \sum_{i \in \pi_g, i \neq m} n_i^2 + (N_g - n_m)^2 \right) - N_g \cdot \left( \sum_{i \in \pi_s, i \neq k_{\pi_s}} n_i^2 + \left( N_s - n_{k_{\pi_s}} \right)^2 \right) \right)}{\left( \mu_e \cdot N_s \cdot N_g{}^2 + \sigma^2 \cdot N_g{}^2 \cdot \left( \sum_{i \in \pi_s, i \neq k_{\pi_s}} n_i^2 + \left( N_s - n_{k_{\pi_s}} \right)^2 \right) \right)^2}.
\end{aligned}
\tag{24}
$$

For $f_1 = N_s \cdot \sum_{i \in \pi_g, i \neq m} n_i^2 - N_g \cdot \sum_{i \in \pi_s, i \neq k_{\pi_s}} n_i^2$ is, since the growth rate of $\sum_{i \in \pi_s, i \neq k_{\pi_s}} n_i^2$ is faster than that of $N_s$, it is reasonable to assume that $f_1$ is decreasing with respect to $N_s$. The minimum value occurs when $\pi_s = \pi_g$, at which point,

$$
\begin{aligned}
f_1 &= \sum_{i \in \pi_g} n_i \left( \sum_{j \in \pi_g} n_j^2 - n_m^2 \right) - \sum_{i \in \pi_g} n_i \left( \sum_{j \in \pi_g} n_j^2 - n_{k_{\pi_s}}^2 \right) \\
&= \sum_{i \in \pi_g} n_i \left[ \left( \sum_{j \in \pi_g} n_j^2 - n_m^2 \right) - \left( \sum_{j \in \pi_g} n_j^2 - n_{k_{\pi_s}}^2 \right) \right] > 0.
\end{aligned}
\tag{25}
$$

Also, $f_2 = N_s \cdot (N_g - n_m)^2 - N_g \cdot \left( N_s - n_{k_{\pi_s}} \right)^2 > 0$:

$$
\begin{aligned}
f_2 &= N_s \cdot (N_g - n_m)^2 - N_g \cdot \left( N_s - n_{k_{\pi_s}} \right)^2 \\
&= N_s N_g (N_g - N_s) - N_g n_{k_{\pi_s}}^2 + N_s n_m^2 + 2 N_g N_s (n_{k_{\pi_s}} - n_m) \\
&= N_g (N_g N_s - N_s^2 - n_{k_{\pi_s}}^2 + n_m^2 + 2 N_s (n_{k_{\pi_s}} - n_m)) \\
&= N_g (\underbrace{N_g N_s - N_s^2}_{>0} + \underbrace{(n_m - N_s)^2 - (n_{k_{\pi_s}} - N_s)^2}_{>0}).
\end{aligned}
\tag{26}
$$

Based on the above derivation, we can conclude that $f'_{RHS,\sigma} > 0$. Therefore, the maximum value of $f_{RHS}$ occurs at the maximum value of $\sigma^2$. Given our assumption that $n_i \leq \frac{\mu_e}{\sigma^2}, \forall i = 1, ..., N$, the maximum of $\sigma^2$ is $\frac{\mu_e}{n_l}, l = \arg \max_{i \in \pi_g} \{n_i\}$. Consequently, the Eq. (23) transforms into,

$$
\begin{aligned}
\lambda &\geq \frac{\mu_e \cdot N_g \cdot N_s{}^2 \cdot n_l + \mu_e \cdot N_s{}^2 \cdot \left( \sum_{i \in \pi_g, i \neq m} n_i^2 + (N_g - n_m)^2 \right)}{\mu_e \cdot N_s \cdot N_g{}^2 \cdot n_l + \mu_e \cdot N_g{}^2 \cdot \left( \sum_{i \in \pi_s, i \neq k_{\pi_s}} n_i^2 + \left( N_s - n_{k_{\pi_s}} \right)^2 \right)} \\
&= \frac{N_s{}^2}{N_g{}^2} \cdot \frac{N_g \cdot n_l + \left( \sum_{i \in \pi_g, i \neq m} n_i^2 + (N_g - n_m)^2 \right)}{N_s \cdot n_l + \left( \sum_{i \in \pi_s, i \neq k_{\pi_s}} n_i^2 + \left( N_s - n_{k_{\pi_s}} \right)^2 \right)}.
\end{aligned}
\tag{27}
$$

Above all, we can determine the optimal egalitarian fairness bound to maintain the core stability of the coalition structure $\pi_g$,

$$
\begin{aligned}
\lambda &\geq \max_{\pi_s \in \pi_g} \left\{ \frac{N_s{}^2}{N_g{}^2} \cdot \frac{N_g \cdot n_l + \left( \sum_{i \in \pi_g, i \neq m} n_i^2 + (N_g - n_m)^2 \right)}{N_s \cdot n_l + \left( \sum_{i \in \pi_s i \neq k_{\pi_s}} n_i^2 + \left( N_s - n_{k_{\pi_s}} \right)^2 \right)} \right\} \\
&= \max_{\pi_s \in \pi_g} \left\{ \frac{N_s{}^2}{N_g{}^2} \cdot \frac{N_g \cdot n_l + d(\pi_g, n_m)}{N_s \cdot n_l + d(\pi_s, n_{k_{\pi_s}})} \right\}.
\end{aligned}
\tag{28}
$$

### A.4.2 Proof of Corollary 2

**Corollary 2** The core-stable grand coalition $\pi_g$, comprising all selfish clients, can asymptotically achieve strict egalitarian fairness, provided that the local dataset sizes of all clients are equal.

**Proof** *Let's consider a grand coalition $\pi_g$ that consists of $M$ clients, and a subset $\pi_s \subset \pi_g$ that consists of $m$ clients. According to Proposition 2, when the local dataset sizes of all clients are equal, the optimal egalitarian fairness that can be achieved is:*

$$
\begin{aligned}
\lambda_{optimal} &= \max_{\pi_s \subset \pi_g} \left\{ \frac{N_s{}^2}{N_g{}^2} \cdot \frac{N_g \cdot n_l + d(\pi_g, n_m)}{N_s \cdot n_l + d(\pi_s, n_{k_{\pi_s}})} \right\} \\
&= \max_{\pi_s \subset \pi_g} \left\{ \frac{m^2 \cdot \left( M + M - 1 + (M - 1)^2 \right)}{M^2 \cdot (m + m - 1 + (m - 1)^2)} \right\} = 1.
\end{aligned}
\tag{29}
$$

*Therefore, it is proven that the grand coalition $\pi_g$ is capable of achieving strict egalitarian fairness.*

### A.4.3 Proof of Proposition 3

**Proposition 3** (Optimal egalitarian fairness under purely welfare altruistic behaviors) Considering all clients act purely welfare altruistic, the grand coalition $\pi_g$ remains core-stable if the achieved egalitarian fairness is bounded by:

$$\lambda \geq \max_{\pi_s \in \pi_g} \left\{ \min \left( \frac{N_s^2}{N_g^2} \cdot \frac{N_g \cdot n_l + d(\pi_g, n_m)}{N_s \cdot n_l + d(\pi_s, f_{\pi_s,1}^{opt})}, \frac{N_c^2}{N_g^2} \cdot \frac{N_g \cdot n_l + d(\pi_g, n_m)}{N_c \cdot n_l + d(\pi_c, f_{\pi_s,2}^{opt})} \right) \right\},$$

*where*

$$k_{\pi_s,1} = \arg\min_{i \in \pi_s} \left\{ \min_{f \in F_i \cap \pi_s} n_f \right\}, k_{\pi_s,2} = \arg\min_{i \in \pi_s} \left\{ \min_{f \in F_i \cap \pi_c} n_f \right\},$$
$$f_{\pi_s,1}^{opt} = \arg\min_{f \in F_{k_{\pi_s,1}} \cap \pi_s} n_f, f_{\pi_s,2}^{opt} = \arg\min_{f \in F_{k_{\pi_s,2}} \cap \pi_c} n_f.$$

(30)

**Proof** *In a context where clients are purely welfare altruistic, the coalition $\pi_g$ maintains core-stable if, for any potential sub-coalition $\pi_s \subset \pi_g$, there is at least one client who prefers $\pi_g$ over $\pi_s$, this means that $\exists i \in \pi_s, u_i(\pi_g) \leq u_i(\pi_s)$, or equivalently,*

$$\exists i \in \pi_s, \max_{f \in F_i} (\{err_f(\pi_g)\}) \leq \max \left( \max_{f \in F_i \cap \pi_s} (\{err_f(\pi_s)\}), \max_{f \in F_i \cap \pi_c} (\{err_f(\pi_c)\}) \right). \quad (31)$$

*where $\pi_c = \pi_g \setminus \pi_s$ is the complement of $\pi_s$. Consequently, to maintain the stability of FL, we can determine the lower bound of $\lambda$ by,*

$$\frac{\max\{err_i(\pi_g)\}_{i=1}^N}{\lambda} \leq \min_{\pi_s \subset \pi_g} \left( \max \left( \max_{i \in \pi_s} \left\{ \max_{f \in F_i \cap \pi_s} err_f(\pi_s) \right\}, \max_{i \in \pi_s} \left\{ \max_{f \in F_i \cap \pi_c} err_f(\pi_c) \right\} \right) \right). \quad (32)$$

*The fairness bound $\lambda$ with respect to a specific $\pi_s$ is,*

$$\frac{err_m(\pi_g)}{\lambda} \leq \max \left( \max_{f \in F_{k_{\pi_s,1}} \cap \pi_s} err_f(\pi_s), \max_{f \in F_{k_{\pi_s,2}} \cap \pi_c} err_f(\pi_c) \right)$$
$$= \max \left( err_{f_{\pi_s,1}^{opt}}(\pi_s), err_{f_{\pi_s,2}^{opt}}(\pi_c) \right),$$

(33)

*where,*

$$m = \arg\max_{i \in \pi_g} \{err_i(\pi_g)\} = \arg\min_{i \in \pi_g} \{n_i\},$$
$$k_{\pi_s,1} = \arg\max_{i \in \pi_s} \{\max_{f \in F_i} err_f(\pi_s)\} = \arg\min_{i \in \pi_s} \{\min_{f \in F_i \cap \pi_s} n_f\},$$
$$k_{\pi_s,2} = \arg\max_{i \in \pi_s} \{\max_{f \in F_i \cap \pi_c} err_f(\pi_c)\} = \arg\min_{i \in \pi_s} \{\min_{f \in F_i \cap \pi_c} n_f\},$$
$$f_{\pi_s,1}^{opt} = \arg\max_{f \in F_{k_{\pi_s,1}} \cap \pi_s} err_f(\pi_s) = \arg\min_{f \in F_{k_{\pi_s,1}} \cap \pi_s} n_f,$$
$$f_{\pi_s,2}^{opt} = \arg\max_{f \in F_{k_{\pi_s,2}} \cap \pi_c} err_f(\pi_c) = \arg\min_{f \in F_{k_{\pi_s,2}} \cap \pi_c} n_f.$$

(34)

*Following the same derivation as in the proof of Proposition 2, the above equation is equivalent to,*

$$\lambda \geq \min \left( \frac{err_m(\pi_g)}{err_{f_{\pi_s,1}^{opt}}(\pi_s)}, \frac{err_m(\pi_g)}{err_{f_{\pi_s,2}^{opt}}(\pi_c)} \right)$$
$$= \min \left( \frac{N_s^2}{N_g^2} \cdot \frac{N_g \cdot n_l + d(\pi_g, n_m)}{N_s \cdot n_l + d(\pi_s, f_{\pi_s,1}^{opt})}, \frac{N_c^2}{N_g^2} \cdot \frac{N_g \cdot n_l + d(\pi_g, n_m)}{N_c \cdot n_l + d(\pi_c, f_{\pi_s,2}^{opt})} \right),$$

(35)

*where*

$$l = \arg\max_{i \in \pi_g} \{n_i\}, N_g = \sum_{i \in \pi_g} n_i, N_s = \sum_{i \in \pi_s} n_i, N_c = \sum_{i \in \pi_c} n_i. \quad (36)$$

*Above all, we can determine the optimal egalitarian fairness bound to maintain the core stability of the coalition structure $\pi_g$,*

$$\lambda \geq \max_{\pi_s \in \pi_g} \left\{ \min \left( \frac{N_s^2}{N_g^2} \cdot \frac{N_g \cdot n_l + d(\pi_g, n_m)}{N_s \cdot n_l + d(\pi_s, f_{\pi_s,1}^{opt})}, \frac{N_c^2}{N_g^2} \cdot \frac{N_g \cdot n_l + d(\pi_g, n_m)}{N_c \cdot n_l + d(\pi_c, f_{\pi_s,2}^{opt})} \right) \right\}. \quad (37)$$

### A.4.4 Proof of Corollary 3

**Corollary 3** The core-stable grand coalition $\pi_g$, consisting of purely welfare clients, can asymptotically achieve strict egalitarian fairness if all clients are friends with the client possessing the smallest dataset size and $N_g \to \infty$.

**Proof** *When all clients are friends with the client possessing the smallest dataset size, it holds that $f_{\pi_s}^{opt} = m$ or $f_{\pi_c}^{opt} = m$. Assume that $f_{\pi_s}^{opt} = m$, Corollary 3 can be proven by mathematical induction as follows. Define the hypothesis $H_0$ as:*

$$H_0: \frac{N_s{}^2}{N_g{}^2} \cdot \frac{N_g \cdot n_l + d(\pi_g, n_m)}{N_s \cdot n_l + d(\pi_s, n_m)} \leq 1, \forall \pi_s \subset \pi_g. \tag{38}$$

*When $\pi_s$ contains only one player $\pi_s = \{n_j\}$, we have,*

$$\frac{n_j{}^2}{N_g{}^2} \cdot \frac{N_g \cdot n_l + d(\pi_g, n_m)}{n_j \cdot n_l + \underbrace{d(\pi_s, n_m)}_{\text{equal to } 0}} \overset{N_g \to \infty}{=} \underbrace{\frac{n_j}{N_g}}_{\text{approaching } 0} + \underbrace{\frac{n_j}{n_l} \cdot \frac{\sum_{i \in \pi_g} n_i^2 - n_m^2}{\left(\sum_{i \in \pi_g} n_i\right)^2}}_{\text{approaching } 0} + \underbrace{\frac{n_j}{n_l} \cdot \frac{(N_g - n_m)^2}{N_g^2}}_{\text{less than } 1} \leq 1. \tag{39}$$

*Assume that the hypothesis $H_0$ holds when $\pi_s$ contains $k$ players, for $\pi_s' = \{\pi_s, n\}$ containing $k+1$ players where the new player has a dataset size of $n$, it holds that,*

$$\begin{aligned}&\frac{N_s'{}^2}{N_g{}^2} \cdot \frac{N_g \cdot n_l + d(\pi_g, n_m)}{N_s' \cdot n_l + d(\pi_s', n_m)} \\ &\leq \frac{N_s{}^2 \cdot (N_g \cdot n_l + d(\pi_g, n_m)) + n^2 \cdot (N_g \cdot n_l + d(\pi_g, n_m)) + 2N_s \cdot n(N_g \cdot n_l + d(\pi_g, n_m))}{N_g{}^2 \cdot (N_s \cdot n_l + d(\pi_s, n_m)) + N_g^2 \cdot (n \cdot n_l + d(\{n\}, n_m)) + 2N_s \cdot n \cdot N_g^2},\end{aligned} \tag{40}$$

*where $N_s' = \sum_{i \in \pi_s'} n_i$. As $N_g \to \infty$, we have,*

$$\frac{2N_s \cdot n \cdot (N_g \cdot n_l + d(\pi_g, n_m))}{2N_s \cdot n \cdot N_g^2} \overset{N_g \to \infty}{=} \frac{n_l}{N_g} + \frac{\sum_{i \in \pi_g} n_i^2 - n_m^2}{N_g^2} + \frac{(N_g - n_m)^2}{N_g^2} \leq 1. \tag{41}$$

*Since $H_0$ holds for any $\pi_s$, the terms in the numerator are each smaller than the corresponding terms in the denominator. Thus, it follows that,*

$$\frac{N_s'{}^2}{N_g{}^2} \cdot \frac{N_g \cdot n_l + d(\pi_g, n_m)}{N_s' \cdot n_l + d(\pi_s', n_m)} \leq 1. \tag{42}$$

*Therefore, the hypothesis $H_0$ is established. A similar mathematical induction process can be applied to $\pi_c$. Thus, $\lambda_{optimal}$ can approach less than 1 under the given conditions, proving the Corollary 3.*

### A.4.5 Proof of Proposition 4

**Proposition 4** (Optimal egalitarian fairness under purely equal altruistic behaviors) Considering all clients act purely equal altruistic, the grand coalition $\pi_g$ remains core-stable if the achieved egalitarian fairness is bounded by:

$$\lambda \geq \max_{\pi_s \in \pi_g} \left( \frac{|F_{k_{\pi_s}}| \cdot N_s{}^2 N_c{}^2}{N_g{}^2} \cdot \frac{N_g \cdot n_l + d(\pi_g, n_m)}{\mathbf{Q}} \right),$$

*where*

$$\begin{aligned}k_{\pi_s} &= \arg\min_{i \in \pi_s} \frac{1}{|F_i|} \left( \sum_{f \in F_i \cap \pi_s} n_f + \sum_{f \in F_i \cap \pi_c} n_f \right), \\ \mathbf{Q} &= N_c^2 \cdot \sum_{f \in F_{k_{\pi_s}} \cap \pi_s} (N_s \cdot n_l + d(\pi_s, n_f)) + N_s^2 \cdot \sum_{f \in F_{k_{\pi_s}} \cap \pi_c} (N_c \cdot n_l + d(\pi_c, n_f)).\end{aligned} \tag{43}$$

**Proof** *In a context where clients are purely equal altruistic, the coalition $\pi_g$ maintains core-stable if, for any potential sub-coalition $\pi_s \subset \pi_g$, there is at least one client who prefers $\pi_g$ over $\pi_s$, this means that $\exists i \in \pi_s, u_i(\pi_g) \leq u_i(\pi_s)$, or equivalently,*

$$\exists i \in \pi_s, \frac{1}{|F_i|} \sum_{f \in F_i} err_f(\pi_g) \leq \frac{1}{|F_i|} \left( \sum_{f \in F_i \cap \pi_s} err_f(\pi_s) + \sum_{f \in F_i \cap \pi_c} err_f(\pi_c) \right), \tag{44}$$

*where $\pi_c = \pi_g \setminus \pi_s$ is the complement of $\pi_s$. Consequently, to maintain the stability of FL, we can determine the lowest fairness bound $\lambda$ by,*

$$\frac{\max\{err_i(\pi_g)\}_{i=1}^N}{\lambda} \leq \min_{\pi_s \subset \pi_g} \left( \max_{i \in \pi_s} \left\{ \frac{1}{|F_i|} \left( \sum_{f \in F_i \cap \pi_s} err_f(\pi_s) + \sum_{f \in F_i \cap \pi_c} err_f(\pi_c) \right) \right\} \right). \tag{45}$$

*The lowest fairness bound $\lambda$ with respect to a specific $\pi_s$ is,*

$$\frac{err_m(\pi_g)}{\lambda} \leq \frac{1}{|F_{k_{\pi_s}}|} \left( \sum_{f \in F_{k_{\pi_s}} \cap \pi_s} err_f(\pi_s) + \sum_{f \in F_{k_{\pi_s}} \cap \pi_c} err_f(\pi_c) \right), \tag{46}$$

*where,*

$$m = \arg\max_{i \in \pi_g} \{err_i(\pi_g)\} = \arg\min_{i \in \pi_g} \{n_i\},$$

$$k_{\pi_s} = \arg\max_{i \in \pi_s} \frac{1}{|F_i|} \left( \sum_{f \in F_i \cap \pi_s} err_f(\pi_s) + \sum_{f \in F_i \cap \pi_c} err_f(\pi_c) \right) \tag{47}$$

$$= \arg\min_{i \in \pi_s} \frac{1}{|F_i|} \left( \sum_{f \in F_i \cap \pi_s} n_f + \sum_{f \in F_i \cap \pi_c} n_f \right).$$

*Following the same derivation as in the proof of Proposition 2, the above equation is equivalent to,*

$$\lambda \geq \frac{|F_{k_{\pi_s}}| \cdot err_m(\pi_g)}{\sum_{f \in F_{k_{\pi_s}} \cap \pi_s} err_f(\pi_s) + \sum_{f \in F_{k_{\pi_s}} \cap \pi_c} err_f(\pi_c)} = \frac{|F_{k_{\pi_s}}| \cdot N_s^{\,2} N_c^{\,2}}{N_g^{\,2}} \cdot \frac{N_g \cdot n_l + d(\pi_g, n_m)}{\mathbf{Q}}, \tag{48}$$

*where,*

$$l = \arg\max_{i \in \pi_g} \{n_i\}, N_g = \sum_{i \in \pi_g} n_i, N_s = \sum_{i \in \pi_s} n_i, N_c = \sum_{i \in \pi_c} n_i,$$

$$\mathbf{Q} = N_c^2 \cdot \sum_{f \in F_{k_{\pi_s}} \cap \pi_s} (N_s \cdot n_l + d(\pi_s, n_f)) + N_s^2 \cdot \sum_{f \in F_{k_{\pi_s}} \cap \pi_c} (N_c \cdot n_l + d(\pi_c, n_f)). \tag{49}$$

*Above all, we can determine the optimal egalitarian fairness bound to maintain the core stability of the coalition structure $\pi_g$,*

$$\lambda \geq \max_{\pi_s \in \pi_g} \left( \frac{|F_{k_{\pi_s}}| \cdot N_s^{\,2} N_c^{\,2}}{N_g^{\,2}} \cdot \frac{N_g \cdot n_l + d(\pi_g, n_m)}{\mathbf{Q}} \right). \tag{50}$$

### A.4.6 Proof of Proposition 5

**Proposition 5** (Optimal egalitarian fairness under friendly welfare altruistic behaviors) Considering all clients act friendly welfare altruistic, the grand coalition $\pi_g$ remains core-stable if the achieved egalitarian fairness is bounded by:

$$\lambda \geq \max_{\pi_s \in \pi_g} \left\{ \min \left( \frac{N_s^{\,2}}{N_g^{\,2}} \cdot \frac{N_g \cdot n_l + d(\pi_g, n_m)}{\mathbf{Q}_1}, \frac{N_s^{\,2} N_c^{\,2}}{N_g^{\,2}} \cdot \frac{N_g \cdot n_l + d(\pi_g, n_m)}{\mathbf{Q}_2} \right) \right\},$$

*where*

$$k_{\pi_s,1} = \arg\min_{i \in \pi_s} \left\{ w \cdot n_i + (1-w) \cdot \min_{f \in F_i \cap \pi_s \cup \{i\}} n_f \right\},$$

$$k_{\pi_s,2} = \arg\min_{i \in \pi_s} \left\{ w \cdot n_i + (1-w) \cdot \min_{f \in F_i \cap \pi_c} n_f \right\},$$

$$f_{\pi_s,1}^{opt} = \arg\min_{f \in F_{k_{\pi_s,1}} \cap \pi_s \cup \{k_{\pi_s,1}\}} n_f, f_{\pi_s,2}^{opt} = \arg\min_{f \in F_{k_{\pi_s,2}} \cap \pi_c} n_f$$

$$\mathbf{Q}_1 = N_s \cdot n_l + w \cdot d(\pi_s, n_{k_{\pi_s,1}}) + (1-w) \cdot d(\pi_s, f_{\pi_s,1}^{opt})$$

$$\mathbf{Q}_2 = N_c^2 \cdot w \cdot \left( N_s \cdot n_l + d(\pi_s, n_{k_{\pi_s,2}}) \right) + N_s^2 \cdot (1-w) \cdot \left( N_c \cdot n_l + d(\pi_c, f_{\pi_s,2}^{opt}) \right). \tag{51}$$

**Proof** *In a context where clients are friendly welfare altruistic, the coalition $\pi_g$ maintains stability if, for any potential sub-coalition $\pi_s \subset \pi_g$, there is at least one client who prefers $\pi_g$ over $\pi_s$, this means that $\exists i \in \pi_s, u_i(\pi_g) \leq u_i(\pi_s)$, or equivalently,*

$$\exists i \in \pi_s, w \cdot err_i(\pi_g) + (1-w) \cdot \max_{f \in F_i \cup \{i\}} (\{err_f(\pi_g)\})$$

$$\leq \max \begin{pmatrix} w \cdot err_i(\pi_s) + (1-w) \cdot \max_{f \in F_i \cap \pi_s \cup \{i\}} (\{err_f(\pi_s)\}), \\ w \cdot err_i(\pi_s) + (1-w) \cdot \max_{f \in F_i \cap \pi_c} (\{err_f(\pi_c)\}) \end{pmatrix}, w \in (0,1), \tag{52}$$

*where $\pi_c = \pi_g \setminus \pi_s$ is the complement of $\pi_s$. Consequently, to maintain the stability of FL, we can determine the lowest fairness bound of $\lambda$ by,*

$$w \cdot \frac{\max\{err_i(\pi_g)\}_{i=1}^N}{\lambda} + (1-w) \cdot \frac{\max\{err_i(\pi_g)\}_{i=1}^N}{\lambda} = \frac{\max\{err_i(\pi_g)\}_{i=1}^N}{\lambda}$$

$$\leq \min_{\pi_s \subset \pi_g} \left( \max \begin{pmatrix} \max_{i \in \pi_s} \left\{ w \cdot err_i(\pi_s) + (1-w) \cdot \max_{f \in F_i \cap \pi_s \cup \{i\}} err_f(\pi_s) \right\}, \\ \max_{i \in \pi_s} \left\{ w \cdot err_i(\pi_s) + (1-w) \cdot \max_{f \in F_i \cap \pi_c} err_f(\pi_c) \right\} \end{pmatrix} \right). \tag{53}$$

*The lowest fairness bound $\lambda$ with respect to a specific $\pi_s$ is,*

$$\frac{err_m(\pi_g)}{\lambda} \leq \max \begin{pmatrix} \max_{i \in \pi_s} \left\{ w \cdot err_i(\pi_s) + (1-w) \cdot \max_{f \in F_i \cap \pi_s \cup \{i\}} err_f(\pi_s) \right\}, \\ \max_{i \in \pi_s} \left\{ w \cdot err_i(\pi_s) + (1-w) \cdot \max_{f \in F_i \cap \pi_c} err_f(\pi_c) \right\} \end{pmatrix}$$

$$= \max \begin{pmatrix} w \cdot err_{k_{\pi_s,1}}(\pi_s) + (1-w) \cdot err_{f_{\pi_s,1}^{opt}}(\pi_s), \\ w \cdot err_{k_{\pi_s,2}}(\pi_s) + (1-w) \cdot err_{f_{\pi_s,2}^{opt}}(\pi_c) \end{pmatrix}, \tag{54}$$

*where,*

$$m = arg\max_{i \in \pi_g} \{err_i(\pi_g)\} = arg\min_{i \in \pi_g} \{n_i\},$$

$$k_{\pi_s,1} = arg\max_{i \in \pi_s} \{w \cdot err_i(\pi_s) + (1-w) \cdot \max_{f \in F_i \cap \pi_s \cup \{i\}} err_f(\pi_s)\}$$

$$= arg\min_{i \in \pi_s} \{w \cdot n_i + (1-w) \cdot \min_{f \in F_i \cap \pi_s \cup \{i\}} n_f\},$$

$$k_{\pi_s,2} = arg\max_{i \in \pi_s} \{w \cdot err_i(\pi_s) + (1-w) \cdot \max_{f \in F_i \cap \pi_c} err_f(\pi_c)\} \tag{55}$$

$$= arg\min_{i \in \pi_s} \{w \cdot n_i + (1-w) \cdot \min_{f \in F_i \cap \pi_c} n_f\},$$

$$f_{\pi_s,1}^{opt} = arg\max_{f \in F_{k_{\pi_s,1}} \cap \pi_s \cup \{k_{\pi_s,1}\}} err_f(\pi_s) = arg\min_{f \in F_{k_{\pi_s,1}} \cap \pi_s \cup \{k_{\pi_s,1}\}} n_f$$

$$f_{\pi_s,2}^{opt} = arg\max_{f \in F_{k_{\pi_s,2}} \cap \pi_c} err_f(\pi_c) = arg\min_{f \in F_{k_{\pi_s,2}} \cap \pi_c} n_f.$$

*Following the same derivation as in the proof of Proposition 2, the above equation is equivalent to,*

$$\lambda \geq \min\left(\frac{err_m(\pi_g)}{w \cdot err_{k_{\pi_s,1}}(\pi_s) + (1-w) \cdot err_{f_{\pi_s,1}^{opt}}(\pi_s)}, \frac{err_m(\pi_g)}{w \cdot err_{k_{\pi_s,2}}(\pi_s) + (1-w) \cdot err_{f_{\pi_s,2}^{opt}}(\pi_c)}\right)$$

$$= \min\left(\frac{N_s^2}{N_g^2} \cdot \frac{N_g \cdot n_l + d(\pi_g, n_m)}{\mathbf{Q}_1}, \frac{N_s^2 N_c^2}{N_g^2} \cdot \frac{N_g \cdot n_l + d(\pi_g, n_m)}{\mathbf{Q}_2}\right), \tag{56}$$

*where,*

$$l = arg\max_{i \in \pi_g} \{n_i\}, N_g = \sum_{i \in \pi_g} n_i, N_s = \sum_{i \in \pi_s} n_i, N_c = \sum_{i \in \pi_c} n_i,$$

$$\mathbf{Q}_1 = N_s \cdot n_l + w \cdot d(\pi_s, n_{k_{\pi_s,1}}) + (1-w) \cdot d(\pi_s, f_{\pi_s,1}^{opt}) \tag{57}$$

$$\mathbf{Q}_2 = N_c^2 \cdot w \cdot (N_s \cdot n_l + d(\pi_s, n_{k_{\pi_s,2}})) + N_s^2 \cdot (1-w) \cdot (N_c \cdot n_l + d(\pi_c, f_{\pi_s,2}^{opt})).$$

*Above all, we can determine the optimal egalitarian fairness bound to maintain the core stability of the coalition structure $\pi_g$,*

$$\lambda \geq \max_{\pi_s \in \pi_g} \left\{\min\left(\frac{N_s^2}{N_g^2} \cdot \frac{N_g \cdot n_l + d(\pi_g, n_m)}{\mathbf{Q}_1}, \frac{N_s^2 N_c^2}{N_g^2} \cdot \frac{N_g \cdot n_l + d(\pi_g, n_m)}{\mathbf{Q}_2}\right)\right\}. \tag{58}$$

### A.4.7 Proof of Proposition 6

**Proposition 6** (Optimal egalitarian fairness under friendly equal altruistic behaviors) Considering all clients act friendly equal altruistic, the grand coalition $\pi_g$ remains core-stable if the achieved egalitarian fairness is bounded by:

$$\lambda \geq \max_{\pi_s \in \pi_g}\left(\frac{(|F_{k_{\pi_s}}|+1) \cdot N_s^2 \cdot N_c^2}{N_g^2} \cdot \frac{N_g \cdot n_l + d(\pi_g, n_m)}{\mathbf{Q}}\right),$$

*where*

$$k_{\pi_s} = arg\min_{i \in \pi_s}\left(w \cdot n_i + (1-w) \cdot \frac{1}{|F_i|+1} \cdot \left(\sum_{f \in F_i \cap \pi_s \cup \{i\}} n_f + \sum_{f \in F_i \cap \pi_c} n_f\right)\right),$$

$$\hat{F}_s = F_{k_{\pi_s}} \cap \pi_s \cup \{k_{\pi_s}\}, \hat{F}_c = F_{k_{\pi_s}} \cap \pi_c,$$

$$\mathbf{Q} = w \cdot (|F_{k_{\pi_s}}|+1) \cdot N_c^2 \cdot (N_s \cdot n_l + d(\pi_s, n_{k_{\pi_s}})) +$$

$$(1-w) \cdot \left(N_c^2 \cdot \sum_{f \in \hat{F}_s} (N_s \cdot n_l + d(\pi_s, n_f)) + N_s^2 \cdot \sum_{f \in \hat{F}_c} (N_c \cdot n_l + d(\pi_c, n_f))\right). \tag{59}$$

**Proof** *(Proposition 6) In a context where clients are friendly equal altruistic, the coalition $\pi_g$ maintains core-stable if, for any potential sub-coalition $\pi_s \subset \pi_g$, there is at least one client who prefers $\pi_g$ over $\pi_s$, this means that $\exists i \in \pi_s, u_i(\pi_g) \leq u_i(\pi_s)$, or equivalently,*

$$\exists i \in \pi_s, w \cdot err_i(\pi_g) + (1-w) \cdot \frac{1}{|F_i|+1} \cdot \sum_{f \in F_i \cup \{i\}} \{err_f(\pi_g)\}$$

$$\leq w \cdot err_i(\pi_s) + (1-w) \cdot \frac{1}{|F_i|+1} \cdot \left(\sum_{f \in F_i \cap \pi_s \cup \{i\}} \{err_f(\pi_s)\} + \sum_{f \in F_i \cap \pi_c} \{err_f(\pi_c)\}\right). \tag{60}$$

*Consequently, to maintain the stability of FL, we can determine the lowest fairness bound of $\lambda$ by,*

$$w \cdot \frac{\max\{err_i(\pi_g)\}_{i=1}^N}{\lambda} + \frac{1-w}{|F_i|+1} \cdot \sum_{f \in F_i \cup \{i\}} \left\{\frac{\max\{err_i(\pi_g)\}_{i=1}^N}{\lambda}\right\} = \frac{\max\{err_i(\pi_g)\}_{i=1}^N}{\lambda} \leq$$

$$\min_{\pi_s \subset \pi_g}\left(\max_{i \in \pi_s}\left\{w \cdot err_i(\pi_s) + \frac{1-w}{|F_i|+1} \cdot \left(\sum_{f \in F_i \cap \pi_s \cup \{i\}} \{err_f(\pi_s)\} + \sum_{f \in F_i \cap \pi_c} \{err_f(\pi_c)\}\right)\right\}\right). \tag{61}$$

*The lowest fairness bound $\lambda$ with respect to a specific $\pi_s$ is,*

$$\frac{err_m(\pi_g)}{\lambda} \leq w \cdot err_{k_{\pi_s}}(\pi_s) + \frac{1-w}{|F_{k_{\pi_s}}|+1} \cdot \left( \sum_{f \in \hat{F}_s} \{err_f(\pi_s)\} + \sum_{f \in \hat{F}_c} \{err_f(\pi_c)\} \right), \quad (62)$$

*where,*

$$m = \arg\max_{i \in \pi_g} \{err_i(\pi_g)\} = \arg\min_{i \in \pi_g} \{n_i\},$$

$$k_{\pi_s} = \arg\max_{i \in \pi_s} \left( w \cdot err_i(\pi_s) + (1-w) \cdot \frac{1}{|F_i|+1} \cdot \left( \sum_{f \in F_i \cap \pi_s \cup \{i\}} \{err_f(\pi_s)\} + \sum_{f \in F_i \cap \pi_c} \{err_f(\pi_c)\} \right) \right)$$

$$= \arg\min_{i \in \pi_s} \left( w \cdot n_i + (1-w) \cdot \frac{1}{|F_i|+1} \cdot \left( \sum_{f \in F_i \cap \pi_s \cup \{i\}} n_f + \sum_{f \in F_i \cap \pi_c} n_f \right) \right),$$

$$\hat{F}_s = F_{k_{\pi_s}} \cap \pi_s \cup \{k_{\pi_s}\}, \hat{F}_c = F_{k_{\pi_s}} \cap \pi_c.$$

$$(63)$$

*Following the same derivation as in the proof of Proposition 2, the above equation is equivalent to,*

$$\lambda \geq \frac{\left(|F_{k_{\pi_s}}|+1\right) \cdot N_s{}^2 \cdot N_c{}^2}{N_g{}^2} \cdot \frac{N_g \cdot n_l + d(\pi_g, n_m)}{\mathbf{Q}}, \quad (64)$$

*where,*

$$l = \arg\max_{i \in \pi_g} \{n_i\}, N_g = \sum_{i \in \pi_g} n_i, N_s = \sum_{i \in \pi_s} n_i, N_c = \sum_{i \in \pi_c} n_i, \quad (65)$$

$$\mathbf{Q} = w \cdot \left(|F_{k_{\pi_s}}|+1\right) \cdot N_c^2 \cdot \left(N_s \cdot n_l + d(\pi_s, n_{k_{\pi_s}})\right) +$$

$$(1-w) \cdot \left( N_c^2 \cdot \sum_{f \in \hat{F}_s} (N_s \cdot n_l + d(\pi_s, n_f)) + N_s^2 \cdot \sum_{f \in \hat{F}_c} (N_c \cdot n_l + d(\pi_c, n_f)) \right). \quad (66)$$

*Above all, we can determine the optimal egalitarian fairness bound to maintain the core stability of the coalition structure $\pi_g$,*

$$\lambda \geq \max_{\pi_s \in \pi_g} \left( \frac{\left(|F_{k_{\pi_s}}|+1\right) \cdot N_s{}^2 \cdot N_c{}^2}{N_g{}^2} \cdot \frac{N_g \cdot n_l + d(\pi_g, n_m)}{\mathbf{Q}} \right). \quad (67)$$

### A.4.8 Fairness Bound Analysis under Generalized-Mean-Form Utility Function

Taking the weighted power-mean welfare function [34] to construct utility function as an example and considering scenarios where clients exhibit friendly equal altruism, the utility of the $i$-th client is:

$$u_i(\pi_g) = \left( \sum_{i=1}^{|F_i|} w_i err_i^q(\pi_g) \right)^{\frac{1}{q}}. \quad (68)$$

The coalition $\pi_g$ remains core-stable if, for any potential sub-coalition $\pi_s \subset \pi_g$, there is at least one client who prefers $\pi_g$ over $\pi_s$. This implies that $\exists i \in \pi_s, u_i(\pi_g) \leq u_i(\pi_s)$, or equivalently,

$$\left( \sum_{i=1}^{|F_i|} w_i err_i^q(\pi_g) \right)^{\frac{1}{q}} \leq \left( w_i \cdot err_i^q(\pi_s) + \sum_{f \in F_i \cap \pi_s} w_f err_f^q(\pi_s) + \sum_{f \in F_i \cap \pi_c} w_f err_f^q(\pi_c) \right)^{\frac{1}{q}}. \quad (69)$$

Consequently, to maintain the stability of FL, we can determine the lowest fairness bound of $\lambda$ by,

$$\left( \sum_{i=1}^{|F_i|} w_i \cdot \frac{\max\left\{err_i^q(\pi_g)\right\}_{i=1}^N}{\lambda} \right)^{\frac{1}{q}} \leq$$

$$\min_{\pi_s \subset \pi_g} \left( \max_{i \in \pi_s} \left( w_i \cdot err_i^q(\pi_s) + \sum_{f \in F_i \cap \pi_s} w_f \cdot err_f^q(\pi_s) + \sum_{f \in F_i \cap \pi_c} w_f \cdot err_f^q(\pi_c) \right)^{\frac{1}{q}} \right). \quad (70)$$

The lowest fairness bound $\lambda$ with respect to a specific $\pi_s$ is,

$$\frac{err_m(\pi_g)}{\lambda} \leq \left( w_{k_{\pi_s}} \cdot err_{k_{\pi_s}}^q(\pi_s) + \sum_{f \in \hat{F}_s} w_f \cdot err_f^q(\pi_s) + \sum_{f \in \hat{F}_c} w_f \cdot err_f^q(\pi_c) \right)^{\frac{1}{q}}, \quad (71)$$

where,
$$\pi_c = \pi_g \setminus \pi_s, m = \arg\min_{i \in \pi_g}\{n_i\},$$
$$k_{\pi_s} = \arg\min_{i \in \pi_s} \left( w_i \cdot n_i^q + \sum_{f \in F_i \cap \pi_s} w_f \cdot n_f^q + \sum_{f \in F_i \cap \pi_c} w_f \cdot n_f^q \right), \tag{72}$$
$$\hat{F}_s = F_{k_{\pi_s}} \cap \pi_s, \hat{F}_c = F_{k_{\pi_s}} \cap \pi_c.$$

Following the same derivation as in the proof of Proposition 2, the above equation is equivalent to,
$$\lambda \geq \left( \frac{N_s^2 \cdot N_c^2}{N_g^2} \right)^q \cdot \frac{(N_g \cdot n_l + d(\pi_g, n_m))^q}{\mathbf{Q}}, \tag{73}$$

where,
$$N_s = \sum_{i \in \pi_s} n_i, N_c = \sum_{i \in \pi_c} n_i, N_g = \sum_{i \in \pi_g} n_i. \tag{74}$$

$$\begin{aligned}
\mathbf{Q} = &\, w_{k_{\pi_s}} \cdot N_c^{2q} \cdot \left( N_s \cdot n_l + d(\pi_s, n_{k_{\pi_s}}) \right)^q \\
&+ N_c^{2q} \cdot \sum_{f \in \hat{F}_s} w_f \cdot (N_s \cdot n_l + d(\pi_s, n_f))^q \\
&+ N_s^{2q} \cdot \sum_{f \in \hat{F}_c} w_f \cdot (N_c \cdot n_l + d(\pi_c, n_f))^q.
\end{aligned} \tag{75}$$

Above all, we can determine the optimal egalitarian fairness bound to maintain the core stability of the coalition structure $\pi_g$,
$$\lambda \geq \max_{\pi_s \in \pi_g} \left( \left( \frac{N_s^2 \cdot N_c^2}{N_g^2} \right)^q \cdot \frac{(N_g \cdot n_l + d(\pi_g, n_m))^q}{\mathbf{Q}} \right). \tag{76}$$

## A.5 Additional experiments

### A.5.1 Heterogeneous client behaviors

Experiments under heterogeneous client behaviors involved configuring the two clients to exhibit purely selfish behavior while the rest were modeled as friendly equal altruistic as in Figure 4. The results verified that our theoretical bound (green dashed line) continued to align with the empirical fairness results (red solid line) under heterogeneous client behaviors.

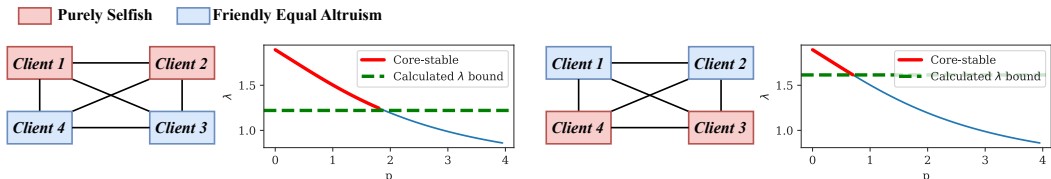

Figure 4: Heterogeneous clients' behaviors: theoretically derived egalitarian fairness bounds (green dashed line) align with empirically achieved egalitarian fairness within the core-stable grand coalition (red solid line).

### A.5.2 Adaptability

Lastly, we examined the adaptability and task-independence of the proposed egalitarian fairness bound through another linear regression task, as illustrated in Figure 5. The experiment underscored that our theoretical bound (green dashed line), independent of task-specific hyperparameters $\theta_i$ or $\epsilon_i$, consistently matched the actual fairness achieved within a core-stable grand coalition (red solid line).

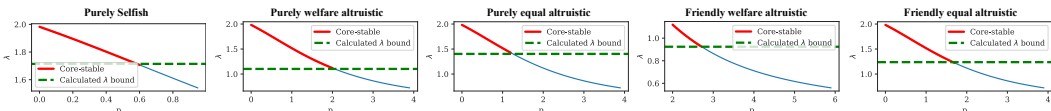

Figure 5: Linear regression: theoretically derived egalitarian fairness bounds (green dashed line) align with empirically achieved egalitarian fairness within the core-stable grand coalition (red solid line) under different client behaviors.

