# OpenReview forum: "Does Egalitarian Fairness Lead to Instability? The Fairness Bounds in Stable Federated Learning Under Altruistic Behaviors"
_NeurIPS.cc/2024/Conference — NeurIPS 2024 poster_

### Official Review · Reviewer_X2VX · 2024-07-05

**Soundness:** 3
**Presentation:** 3
**Contribution:** 1
**Rating:** 5
**Confidence:** 4

**Summary:**

The paper builds on a previous model of FL to study trade-offs between core stability and egalitarian fairness when agents exhibit altruistic behaviors. The authors provide egalitarian fairness bounds under core stability solutions across different cases of altruism. There is also a small experimental part where the authors verify their theoretical results over heterogeneous altruistic behaviors as well.

**Strengths:**

- The idea of introducing the altruistic behavior of the agents in an FL setting is very interesting and well-motivated.

- The modeling of the altruistic behaviors is very clear and reasonable.

**Weaknesses:**

- I am not sure if the authors adequately address their intended question: Does Egalitarian Fairness Lead to Instability? As I read the text, I expected the authors to design egalitarian fair solutions and assess their stability. However, they focus on explaining how high levels of egalitarian fairness are expected under core stability.

- In my view, the most important results are in Section 5.2. However, I am not sure how significant and interesting these findings are. It is clear that there is a trade-off between core stability and egalitarian welfare since they can be seen as two different fairness constraints with quite different goals. So, it is quite expected that we cannot have perfect egalitarian fairness when the goal is to ensure that the grand coalition is stable. Moreover, regarding the bounds of lambda, which are these long expressions, I am quite unsure how intuitive and helpful they can be.

- The findings in Section 5.1 are not surprising at all. When the utility function of the agents changes, it is expected that a core stable solution will change, and different egalitarian welfare levels will be achieved under different solutions.

- The experiments are conducted with an extremely small number of agents, namely 4. I would expect experiments with at least, say, 15 agents for the results to be more interesting.

**Questions:**

- What do we learn from the experiments? What is the take away message?

- Are the bounds that you provide in Section 5.2 tight?

**Limitations:**

No limitations

---

> ### Author Rebuttal · Authors · 2024-08-04
>
> We thank the reviewer for finding our work interesting and well-motivated. We also appreciate the detailed comments posed by the reviewer. Please find below the point-to-point responses to the reviewer's comments.
>
> > **W1 (Question addressing):**
>
> We'd like to clarify how the obtained egalitarian fairness bounds address our original question: *Does Egalitarian Fairness Lead to Instability?*
>
> - We derives the lower bounds of achievable egalitarian fairness under different client behaviors for an FL system to maintain core stability. These bounds clarify that   **"egalitarian fairness leads to instability" happens only when a model trainer prioritizes fairness below the derived fairness bounds**, answering the original question with **tight theoretical support**.
>
>
>
> We  develop egalitarian fairness bounds for stable FL systems **instead of** designing egalitarian fair solutions and assessing their stability for:
>
> -  Rationality: Core stability is a prerequisite for the healthy operation of FL systems. As a result, we investigate the egalitarian fairness bounds under core stability.
>
> -  Tightness：Designing egalitarian fair solutions and assessing their stability does not provide a tight solution about egalitarian fairness bound.  As $\lambda$ is a continuous variable, it does not yield a tight solution to adjust the achieved egalitarian fairness $\lambda$ and monitor when stability is compromised.
>
>
> > **W2 (Findings in Section 5.2):**
>
> Thank you for your comment. The **significance** of Section 5.2 lies in offering **tightly theoretical support** to clarify the misconception that "egalitarian fairness leads to instability in FL" in previous work **rather than relying on intuition**.
>
> Specifically, Section 5.2 theoretically analyzes the achievable egalitarian fairness bounds under **various client behavior scenarios**, providing insights into **establishing appropriate egalitarian fairness** in fair FLs. Additionally, we discuss how to extend the calculation of fairness bounds to accommodate clients with **heterogeneous behaviors** within the coalition.
>
>
> Under behaviors such as equal altruistic or friendly altruistic, the complexity of the egalitarian fairness bounds increases because each individual must consider multi-related clients, which introduces additional variables into the calculations.
>
> > **W3 (Findings in Section 5.1):**
>
> Thank you for your comment. We provide Section 5.1 to **highlight** that clients' altruistic behaviors and friendship relationships affect optimal egalitarian fairness in a core-stable coalition and **empirically refute** the misconception that "egalitarian fairness leads to instability."
>
> > **W4 (More clients):**
>
>  Following your suggestion, we add experiments on more clients. The results below verify that the egalitarian fairness bounds in a stable FL system are influenced by varying client behaviors and friendship networks and verify that the calculated egalitarian fairness bounds are closely aligned with the actual results (minor deviations exist due to discretization errors from adjustments in the fairness constraint parameter, $p$).
>
> **Tab. 1 Calculated egalitarian fairness bound and Actual achieved egalitarian fairness under fully-connected**
> |        | Selfish | Purely Welfare Altruism | Friendly Welfare Altruism | Purely Equal Altruism | Friendly Equal Altruism | 6 Selfish + others Friendly Equal Altruism |
> |--------------------------------------|---------|--------------------------|----------------------------|------------------------|--------------------------|-----------------------------------------|
> | **10 Clients [10,20,30,40,50,60,70,80,90,100]**  |
> | Calculated egalitarian fairness bound | 1.27    | 1.02     | 1.04      | 1.16   | 1.17     | 1.27   |
> | Actual achieved egalitarian fairness  | 1.27    | 1.03     | 1.05   | 1.16   | 1.17     | 1.27  |
> | **15 clients [10,10,30,30,40,40,50,50,60,60,70,80,90,100,100]**   |
> | Calculated egalitarian fairness bound | 1.18    | 0.99 | 1.04   | 1.10   | 1.12     | 1.13   |
> | Actual achieved egalitarian fairness  | 1.18    | 1.00 | 1.04   | 1.10   | 1.13     | 1.14   |
>
> **Tab. 2 Calculated egalitarian fairness bound and Actual achieved egalitarian fairness under partially-connected**
> |        | Selfish | Purely Welfare Altruism | Friendly Welfare Altruism | Purely Equal Altruism | Friendly Equal Altruism | 6 Selfish + others Friendly Equal Altruism |
> |--------------------------------------|---------|--------------------------|----------------------------|------------------------|--------------------------|-----------------------------------------|
> | **0 - 2 - 4 - 6 - 8 - 0, 1 - 3 - 5 -7 - 9 - 1**           |
> | Calculated egalitarian fairness bound | 1.27    | 1.08     | 1.07      | 1.20   | 1.18     | 1.27   |
> | Actual achieved egalitarian fairness  | 1.27    | 1.09     | 1.07       | 1.20   | 1.19     | 1.27  |
> | **0 - 2 - 4 - 6 - 8 - 10 - 12 - 14 - 0, 1 - 3 - 5 -7 -9 -11 - 13 - 1**   |
> | Calculated egalitarian fairness bound | 1.18    | 1.04     | 1.04  | 1.13   | 1.12     | 1.18   |
> | Actual achieved egalitarian fairness  | 1.18    | 1.04     | 1.05   | 1.13  | 1.12     | 1.18   |
>
> >**Q1 (Experiments):**
>
> Thank you for your question. Our experiments highlight two key findings:
> - The theoretically derived achievable egalitarian fairness bounds align closely with the fairness achieved in a stable FL system, confirming their **tightness**.
> - The **adaptability** of these bounds is further validated by Figure 5, which compares the theoretical and actual egalitarian fairness in an additional linear regression task.
>
> >**Q2 (Tightness):**
>
> Thank you for your question. The theoretical bounds are **tight**, which are rigorously derived under the framework set by Lemma 1 and adhere to the conditions for core stability specified in Lemma 2 to 4.

---

> > ### Comment · Reviewer_X2VX · 2024-08-09
> > **No further questions**
> >
> > I would like to thank the authors for their answer and the clarifications. I do not have any further questions at this point. I increase my score based on the new experiments and the further explanation regarding the scope of the paper.

---

### Official Review · Reviewer_r5t6 · 2024-07-08

**Soundness:** 3
**Presentation:** 3
**Contribution:** 3
**Rating:** 5
**Confidence:** 3

**Summary:**

The authors explored the impact of egalitarian fairness on the stability of Federated Learning (FL) systems. FL allows multiple clients to train a global model collaboratively without sharing their local data, thus preserving privacy. The authors addressed the concern that achieving egalitarian fairness, which aims to ensure uniform model performance across clients, might destabilize the FL system by prompting data-rich clients to leave. The authors modeled FL systems as altruism coalition formation games (ACFGs) and propose theoretical bounds for egalitarian fairness that maintain core stability under various types of altruistic behaviors. They disproved the notion that egalitarian fairness necessarily leads to instability and provided experimental validation of their theoretical bounds.

**Strengths:**

1. The authors presented a novel perspective by integrating concepts from game theory and social choice theory to analyze the stability of FL systems under the influence of egalitarian fairness.
2. The theoretical contributions are robust and well-supported by rigorous mathematical proofs. The proposed fairness bounds provide valuable insights into maintaining stability in FL systems.
3. The paper is well-organized, with clear definitions and logical progression of ideas. The use of examples and detailed explanations helps in understanding the complex concepts.

**Weaknesses:**

1. The experiments conducted to validate the theoretical bounds are limited to relatively simple tasks, such as mean estimation and linear regression with a fixed number of clients. While these experiments demonstrate alignment with the theoretical results, extending the validation to more complex and diverse FL scenarios would strengthen the empirical evidence.
2. The sensitivity of the proposed fairness bounds to variations in client behavior and network topology is not fully explored. While the authors provided insights into different types of altruistic behaviors and their impact on stability, a more detailed analysis of how changes in client behavior or the structure of the friends-relationship network affect the theoretical bounds would add depth to the findings.
3. The use of clients' and client's is not rigorous enough (such as Line 22 and Line 128) and needs to be carefully checked.

**Questions:**

1. How do the proposed fairness bounds perform in more complex FL scenarios?
2. How sensitive are the theoretical bounds to variations in client behavior and network topology? Are there specific conditions under which the bounds might fail to ensure stability?

**Limitations:**

The authors have adequately addressed the limitations and potential negative societal impacts of their work in Section 7. They have acknowledged the influence of clients' altruistic behaviors and the configuration of the friend-relationship network on the stability and fairness within federated learning (FL) coalitions. They also identified potential limitations such as the impact of proportional fairness and the necessity to design suitable incentive mechanisms to retain clients when high egalitarian fairness is mandatory.

---

> ### Author Rebuttal · Authors · 2024-08-04
>
> We are delighted that the reviewer found our work novel and valuable, with robust theoretical support.
> Thank you for your positive opinions and insightful comments.
>
> > **W1 & Q1 (More complex and diverse FL scenarios):**
>
> Thank you for your comment.  Following your suggestion, we first add more experiments on more complex scenarios, as in Tab.1 and Tab.2. The results confirm that the calculated egalitarian fairness bounds are closely aligned with the actual achieved fairness values (minor deviations due to discretization errors from adjustments in the fairness constraint parameter, $p$).
>
>
> **Tab. 1 Calculated egalitarian fairness bound and Actual achieved egalitarian fairness under fully-connected**
> |        | Selfish | Purely Welfare Altruism | Friendly Welfare Altruism | Purely Equal Altruism | Friendly Equal Altruism | 6 Selfish + others Friendly Equal Altruism |
> |--------------------------------------|---------|--------------------------|----------------------------|------------------------|--------------------------|-----------------------------------------|
> | **10 Clients [10,20,30,40,50,60,70,80,90,100]**           |
> | Calculated egalitarian fairness bound | 1.27    | 1.02    | 1.04     | 1.16   | 1.17     | 1.27   |
> | Actual achieved egalitarian fairness  | 1.27    | 1.03     | 1.05   | 1.16   | 1.17     | 1.27  |
> | **15 clients [10,10,30,30,40,40,50,50,60,60,70,80,90,100,100]**    |
> | Calculated egalitarian fairness bound | 1.18    | 0.99     | 1.04  | 1.10   | 1.12     | 1.13   |
> | Actual achieved egalitarian fairness  | 1.18    | 1.00     | 1.04 | 1.10   | 1.13     | 1.14   |
>
> **Tab. 2 Calculated egalitarian fairness bound and Actual achieved egalitarian fairness under partially-connected**
> |        | Selfish | Purely Welfare Altruism | Friendly Welfare Altruism | Purely Equal Altruism | Friendly Equal Altruism | 6 Selfish + others Friendly Equal Altruism |
> |--------------------------------------|---------|--------------------------|----------------------------|------------------------|--------------------------|-----------------------------------------|
> | **0 - 2 - 4 - 6 - 8 - 0, 1 - 3 - 5 -7 - 9 - 1**           |
> | Calculated egalitarian fairness bound | 1.27    | 1.08     | 1.07      | 1.20   | 1.18     | 1.27   |
> | Actual achieved egalitarian fairness  | 1.27    | 1.09     | 1.07       | 1.20   | 1.19     | 1.27  |
> | **0 - 2 - 4 - 6 - 8 - 10 - 12 - 14 - 0, 1 - 3 - 5 -7 -9 -11 - 13 - 1**           |
> | Calculated egalitarian fairness bound | 1.18    | 1.04     | 1.04       | 1.13   | 1.12     | 1.18   |
> | Actual achieved egalitarian fairness  | 1.18    | 1.04     | 1.05       | 1.13  | 1.12     | 1.18   |
>
> We'd also like to clarify why we currently use mean estimation and linear regression for our experiments: the error in these tasks is exact and can be expressed in closed form, which allows us to do the following **tight** theoretical analysis. For more complex tasks, such as neural network training, the error is uncertain and influenced by model parameters, training methods, and specific client data distributions.  However, the suggested point is well taken:    We acknowledge that our paper focuses on FL tasks with errors exactly  enough to facilitate rigorous theoretical analysis. We explained this point in Section 3, **Lines 131-133**, and gave this limitation in **Lines 349-351**. We would like to revise the paper, particularly the abstract and introduction, to further clarify that "we use the **stylized** model to generate the following **insights** in fairness."
>
>
> >**W2 & Q2  (Sensitivity):**
>
> Thank you for your comment. Based on Eq. (3) to (7), we'd like to analyze how the theoretical bounds are sensitive to client behavior and network topology.  Taking the fairness bound under purely welfare altruistic behaviors in Eq. (4) as an example,  the set of friends $F_i$ for each client $i$ changes with different network topologies, resulting the variance on parameters **$f^{opt} _{\pi_s,1}$, $f^{opt} _{\pi_s,2}$** (variables that identify the client with the smallest data volume within the network topologies of each sub-coalition), thereby leading to different theoretical bounds.
> Following the same settings as the mean estimation task in Section 6, we constructed three types of network topologies:
>
> - T1: fully connected;
>
> - T2: 0 (20), 1 (40) - 2 (50) - 3 (100) - 1 (40) , where only clients 1, 2, and 3 are connected;
>
> -  T3: 0 (20), 1 (40), 2 (50) - 3 (100), where only clients 2 and 3 are connected.
>
> According to our analysis, the values of  $f^{opt} _{\pi_s,1}$, $f^{opt} _{\pi_s,2}$ will gradually increase from T1 to T3, leading to a decrease in the denominator value in the theoretical bounds. Consequently, the theoretical bounds are expected to increase.
> The experimental results, as shown in the table below, are **consistent** with our expectations. The analysis and experiments demonstrate that the theoretical bound calculation is **effective** under varying client behavior and network topology.
>
>
>
>
> |        |  Purely Welfare Altruism | Friendly Welfare Altruism | Purely Equal Altruism | Friendly Equal Altruism |
> |---------|--------------------------|----------------------------|------------------------|--------------------------|
> | **fully-connected**           |
> | Calculated egalitarian fairness bound |  1.08| 0.98| 1.30| 1.23|
> | Actual achieved egalitarian fairness  |  1.08    |0.98| 1.31| 1.24|
> | **0,1-2-3-1** |
> | Calculated egalitarian fairness bound |  1.12     | 1.16| 1.40| 1.32|
> | Actual achieved egalitarian fairness  |  1.12     | 1.16| 1.41| 1.33|
> | **0, 1 , 2-3**  |
> | Calculated egalitarian fairness bound |  1.91 ↑    | 1.29 ↑ | 1.91 ↑| 1.43  ↑|
> | Actual achieved egalitarian fairness  |  1.91 ↑      | 1.30  ↑| 1.91 ↑| 1.44   ↑|
>
> > **W3 (Check the use of "clients'" and "client's"):**
>
> Thank you for your comment. We have checked the use of 'clients'' and 'client's' throughout the paper and verified their logical and grammatical correctness.

---

### Official Review · Reviewer_CtFP · 2024-07-17

**Soundness:** 4
**Presentation:** 3
**Contribution:** 2
**Rating:** 6
**Confidence:** 4

**Summary:**

This paper examines the relationship between egalitarian fairness concepts and stability in federated learning, where multiple clients collaboratively train a shared model while retaining local data privacy. Egalitarian fairness promotes uniform model performance across clients, but this can reduce performance for data-rich clients, potentially causing instability, as these clients might leave for better-performing coalitions. This work employs cooperative game theory and social choice to frame FL systems as altruism coalition formation games, suggesting that core instability issues are linked to clients' altruism and their network of relationships. The authors propose optimal egalitarian fairness bounds that maintain core stability under various altruistic behaviors, suggesting that egalitarian fairness does not necessarily lead to instability.

**Strengths:**

The main strength of this paper is its comprehensive approach to analyzing federated learning under both selfish and altruistic behavior. By considering both the performance of machine learners and the game-theoretic aspects of how federated learners interact, the authors provide a well-rounded analysis of the relationship between egalitarian fairness and stability. The use of game theory and social choice theory to frame FL systems as altruism coalition formation games is particularly innovative, linking instability issues to clients' altruism and their network of relationships. The proposed optimal egalitarian fairness bounds that maintain core stability under various altruistic behaviors are a significant theoretical advancement, disproving the assumption that egalitarian fairness inevitably leads to instability. The technical correctness of the results, subject to the model assumed, further solidifies the paper's contribution to advancing the state of the art in this field. Experimental validation then convincingly supports these theoretical findings with empirical outcomes.

**Weaknesses:**

I'm left with some questions about the motivations for the model, though the authors do a good job motivating it with prior work. I'm also left wondering whether some results and assumptions can be generalized, as the analysis is somewhat rigid; the paper would be much stronger if some results held in greater generality. I also question the model of FL used here (FedAvg), as it simply averages over model parameters (it would be nice to see models where clients submit gradient updates rather than complete models, and also where the central authority itself optimizes for egalitarian or other fairness objectives, rather than averaging). Finally, I argue the client utilities don't fully reflect the impact of overfitting.

**Questions:**

Clearly the FedAvg rule for $\theta$ (line 126) promotes privacy in some sense, but with modest added noise, is the mechanism differentially private?

I would argue that definition 1 is more closely related to the demographic parity concept of equalized error [10], not egalitarian fairness. I would claim that in loss contexts, egalitarian welfare (or more aptly, malfare) is simply the max over average losses of agents [8], but inequality between agents is not antithetical to egalitarian fairness, rather egalitarianism is indifferent to the performance of high performing groups so long as low performing groups cannot be improved. I suppose this is more of a terminology issue, though I do ask if an additive version of this could be explored, where error values are compared to the maximum per group error.

Lemma 1 is quite confusing to me. $\mu_{e}$ depends on i, but notation does not reflect this. By expected value of the variance, you seem to mean expected value of the raw variance, but it's not clear why I care about that quantity? Is this assuming realizability, and therefore the square here corresponds to square loss? Is the method specific to square error though? Why not work directly in terms of loss? These inferences are somewhat confirmed in section 6, though I suspect a method is not really specific to square error, and I claim a more general description in terms of generic loss and loss variance would be easier to follow.

I also think this result does not properly consider overfitting, so I don't think it's very well motivated in a machine learning context. Loss variances should be model (parameter) dependent, so if this is in terms of minimum variance, I suspect it's a reasonable lower bound, and if maximum variance is considered (and multiplied by the log of the model class cardinality or some similar capacity measure), it may be a reasonable upper bound? It's also not totally clear to me which quantities refer to distributions and which refer to data sets in this definition. I would be interested to see a more general analysis, in terms of quantities like Rademacher averages that better characterize overfitting in machine learning [1,2,5]. Some discussion of how this applies to fair learning in particular would be appreciated, especially as works like [9] show that the process of optimizing fair ML objectives has a regularizing effect and can actually reduce overfitting.

In lemmas 2,3,&4, the requirement is that this holds for all i, correct? These are a bit tricky to read, because $\mu_{e}$ depends on i, although $\sigma^{2}$ does not, and the notation does not reflect this. Moreover, it seems unfortunate that we reach the same conclusion in all three cases, is this necessary (i.e., are there counterexamples to improvement in each?).

I think these results can only be true if client distributions are drawn IID. This should be clarified in lemma 1. This is a big assumption, since I might expect friends’ distributions to be correlated (which incentivizes them to deviate, harming core stability). But I also need to ask, is the variance between distributions the sample variance or the distribution variance? I think this matters a lot, since presumably there are relatively few agents, and if the sample variance is what matters, then why can't we condition on the sample (in which case independence wouldn’t matter)?


In section 4.1, I think it would be nice to generalize this idea to define a client's utility function as some welfare function of their own value and the value of their friends. I think this would simplify the presentation, but also you would be able to use more sophisticated concepts, like weighted power-means.

Moreover, since the same result holds for purely altruistic and friendly altruistic cases, this immediately implies that the client utility functions can be the utilitarian maximum social welfare (malfare) function [6,11], and some set of weighted utilitarian welfare (malfare) functions. I suspect it could easily generalize to any weighted utilitarian welfare function. I would also be very interested to see if these results held for any unweighted power-mean welfare function, any weighted power-mean welfare function, or any Gini malfare function [6], as they do in some sense lie between welfare altruism and equal altruism, though I don't think these results follow directly from your stated lemmas.



Minor points:

149 “satisfy” should be “satisfies.” Moreover this definition feels a bit redundant with (1).

158 It seems strange to mix a sociological definition of friend with a mathematical definition. Moreover, while I am not a sociologist, it seems wrong that “friend” would be the most intimate trusted voluntary category. If so, what is a best friend, and where does that leave a partner? But is this discussion even necessary?

Perhaps definitions 2 3 and 4 can be merged to describe the coalitional game?

163 “does not exist” to “does not exist a”


171 I dislike this terminology. I would describe both types of altruism as considering different types of the welfare among friends, but I would term them “egalitarian altruism” and “utilitarian altruism,” respectively [1].

**Limitations:**

FedAvg

I don't love the FedAvg aggregation rule of line 126 (though I appreciate that the authors analyze an established model). This seems appropriate when groups train relatively similar models on their own, and the lost surface is relatively smooth. I suspect this is a decent rule when each client has the same optimal model, but due to limited sampling wouldn't identify this from their own data.

The approach seems particularly problematic for non-convex loss surfaces, or models with symmetry (e.g., neural networks or mixture models where averaging the parameters of two identical models can produce a completely different model). I’d like to see more discussion of these limitations (or generalization of the model).

However, I would like to see some consideration of other models of FL, for instance when clients send gradient updates, rather than raw model parameters. From reading the introduction, I expected the FL to be trained to incentivize egalitarian fairness, but this simple averaging rule is neither utilitarian optimal, nor is it egalitarian optimal. Works like [3] consider sampling implications for this, and biased SGD analyses [4,7] also seem appropriate to discuss, at least in related work.

Overfitting

I would also like to see some consideration of overfitting. It seems tautological that nonaltruistic agents would prefer to train their own model when only training loss is considered, and while altruism may incentivize them to share their data, there is also the non-altruistic effect of reduced overfitting. It seems this would also disincentivize the formations of small coalitions, so I would be very interested to see what impact these factors have on the work.


References:

[1] An axiomatic theory of provably-fair welfare-centric machine learning
C Cousins
Advances in Neural Information Processing Systems 34, 16610-16621

[2] Revisiting fair-PAC learning and the axioms of cardinal welfare
C Cousins
International Conference on Artificial Intelligence and Statistics, 6422-6442

[3] Jacob D Abernethy, Pranjal Awasthi, Matthäus Kleindessner, Jamie Morgenstern, Chris Russell, and Jie Zhang. Active sampling for min-max fairness. In International
Conference on Machine Learning, volume 162, 2022.

[4] A guide through the zoo of biased SGD
Y Demidovich, G Malinovsky… - Advances in Neural …, 2024 - proceedings.neurips.cc,

[5] Uncertainty and the social planner’s problem: Why sample complexity matters
C Cousins
Proceedings of the 2022 ACM Conference on Fairness, Accountability, and …

[6] Algorithms and Analysis for Optimizing Robust Objectives in Fair Machine Learning
Cyrus Cousins

[7] Hu, Yifan, et al. "Biased stochastic first-order methods for conditional stochastic optimization and applications in meta learning." Advances in Neural Information Processing Systems 33 (2020): 2759-2770.

[8] John Rawls. A theory of justice. Harvard University Press, 1971.

[9] Cousins, Cyrus, I. Elizabeth Kumar, and Suresh Venkatasubramanian. "To Pool or Not To Pool: Analyzing the Regularizing Effects of Group-Fair Training on Shared Models." International Conference on Artificial Intelligence and Statistics. PMLR, 2024.

[10] Dwork, Cynthia, et al. "Fairness through awareness." Proceedings of the 3rd innovations in theoretical computer science conference. 2012.

[11] Deschamps, R., Gevers, L.: Leximin and utilitarian rules: A joint characterization. Journal of Economic Theory 17(2), 143–163 (1978)

---

> ### Author Rebuttal · Authors · 2024-08-04
>
> We are delighted that the reviewer found our motivations and  methods innovative and significant  towards an important research question. Thank you for your positive opinions and insightful comments.
>
> **Weakness:** Thank you for your thoughtful comments. We provided responses to the points in Q4 and L2 (Overfitting), Q7 and  Q8 (Welfare function), L1 (FedAvg).
>
> >**Q1 (Privacy):**
>
> FedAvg can provide differential privacy with modest added noise, i.e., DP-FedAvg [1] adding Gaussian noise to the final averaged update to ensure privacy.
>
> [1] McMahan, H. Brendan, et al. "Learning Differentially Private Recurrent Language Models." ICLR. 2018.
>
> >**Q2 (Terminology):**
>
> The distinction between egalitarian fairness and demographic parity lies in the focus on fairness, whether it's between individual clients or between protected groups defined by sensitive attributes. In FL,  improving the model's performance on some clients inevitably reduces performance on others; thus, minimizing the maximum loss over an average of agents becomes a sufficient condition for equalized error among clients and vice versa. Therefore, we adopt equalized error among clients to quantify egalitarian fairness, consistent with [2].
>
> [2] Kate Donahue and Jon Kleinberg. Fairness in model-sharing games. WWW '23. 2023.
>
> >**Q3 (Variables in Lemma 1):**
>
> The notation $\mu_e$ does not specify a particular $i$ because, according to Lemma 1, the mean value $\theta_i$ and standard deviation $\epsilon_i$ of each client's dataset $\mathcal D_i$ is i.i.d. sampled from $\Theta$, resulting in a **consistent**  $\mu_e$ across clients. Additionally, the error in Eq. (1) is derived from $err_i=\left(\boldsymbol{x}^{T} \hat{\boldsymbol{\theta}}_{i}-y\right)^{2}$,  which is actually a generic loss.
>
>
> >**Q4 and L2 (Overfitting):**
>
> Overfitting in machine learning leads to uncertainty in the error outlined in Eq. (1), which is influenced by model structure and parameters, choice of training algorithms, and the specific data distributions of clients, etc.  Centering on exploring the relationship between egalitarian fairness and stability, we use the stylized model in Lemma 1, which provides a closed-form error to derive precise relations and generate insights into fairness settings in fair FLs.  We'd like to revise Lines 349-351's discussion by clarifying the overfitting challenges in more generalized machine learning scenarios. Thank you for highlighting this point!
>
> >**Q5 (Lemmas 2,3,&4):**
>
> The requirement $n_i\le \frac{\mu_e}{\sigma^2}$ holds for all $i=1,...,N$, as explained in Q3 that $\mu_e$ is consistent across clients. Lemmas 2–4 provide sufficient conditions for core stability across these behaviors. This setup ensures that each lemma is tailored to specific scenarios, supporting the tightness of our subsequent propositions.
>
> > **Q6 (IID and Variance):**
>
> Clients are **heterogeneous**  as they have different parameters governing their data distribution and different
> amounts of data they have noisily drawn from their own distribution in Lemma 1.  Regarding your second question, the mentioned variance refers to **distribution variance** rather than sample variance.
>
> > **Q7 and  Q8 (Welfare function):**
>
>  We'd like to clarify how the key ideas in our work **extend** to more utility functions. Taking the weighted power-mean welfare function to construct utility functions as an example and considering scenarios where clients exhibit friendly equal altruism, the utility of the  $i$-th client is:
> $u_i(\pi_g) = \left( \sum_{i=1}^{|F_i|} w_i err_i^q(\pi_g) \right)^{\frac{1}{q}}$.
>
> The coalition $\pi_g$ remains core-stable if,
> $\exists i\in \pi_s,\left( \sum_{i=1}^{|F_i|} w_i err_i^q(\pi_g) \right)^{\frac{1}{q}} \le
> \left(w_i \cdot err_i^q(\pi_s)+\sum_{f \in  F_i \cap \pi_s} w_f err_f^q(\pi_s)+\sum_{f \in  F_i \cap \pi_c} w_f err_f^q(\pi_c)  \right)^{\frac{1}{q}}$.
>
> Following a similar process in Appendix Eq. (16)-(27),
> we have $\lambda \ge  \frac{\left ( {N_s}^2\cdot{N_c}^{2}\cdot (N_g\cdot n_l+d(\pi_g,n_m)) \right )^q }{{N_g}^{2q}\left ( w_{k_{\pi_s}}\cdot N_c^{2q}\cdot \left (N_s\cdot n_l+d(\pi_s,n_{k_{\pi_s}}) \right )^q  + N_c^{2q}\cdot \underset{f\in \hat{F_s}}{\sum} w_f\cdot  \left (  N_s\cdot n_l+d(\pi_s,n_{f})   \right )^q
>   +N_s^{2q}\cdot \underset{f\in \hat{F_c} }{\sum} w_f\cdot \left ( N_c\cdot n_l+d\left ( \pi_c,n_f \right )\right )^q\right ) }$.
> When setting weights w=[0.7,0.1,0.1,0.1] and $q$=1.2, the actually achieved egalitarian fairness was **1.45**, aligned with the calculated fairness bound of **1.44**. Minor deviations exist due to discretization errors of $p$ in Line 312.
>
> > **Minor points (Sociological definition and  terminology):**
>
> Thank you for pointing out the typo errors. Regarding "friend" in Definition 3, we attempt to draw an analogy from sociological friendships to describe how "friends" are in FL. We think **removing** the sociological definition of "friend" will be better. Examples of "friendship" in FL, such as  Lines 47-49, are enough for readers to better understand the "friend " in FL. We agree that "**utilitarian**" better reflects the focus on the worst-off client than "welfare" and revise it following your suggestion. Thank you for your suggestion!
>
> > **L1 (FedAvg):**
>
> Regarding the expansion into broader FL models, our mean estimation task model, adapted from Donahue & Kleinberg, serves as a prototypical task for analyzing the stability of federating coalitions. However, the broader point is well-taken: We should clarify that our paper is investigating an FL model with errors exactly  enough to allow for **tight** theoretical analysis. We explained this point in Section 3, **Lines 131-133**, and gave this limitation in **Lines 349-351**. We would like to revise the paper, particularly the abstract and introduction, to further clarify that "we use a **stylized** model to generate the following **insights** in fairness."
>
> > **L2 (Overfitting):**
>
> Please refer to Q4 and L2 (Overfitting).

---

### Official Review · Reviewer_LduT · 2024-07-18

**Soundness:** 4
**Presentation:** 3
**Contribution:** 4
**Rating:** 8
**Confidence:** 5

**Summary:**

This paper rigorously answered a critical question regarding fair federated learning: Does egalitarian fairness lead to instability? It analyzed and presented the influence of clients’ altruistic behaviors and the configuration of the friend-relationship network on the achievable egalitarian fairness within a core-stable federated learning (FL) coalition.

The optimal egalitarian fairness bounds without compromising core stability are explored theoretically. These novel insights can be leveraged to establish appropriate egalitarian fairness in improving the alignment of FL processes with societal welfare and ethical standards.

**Strengths:**

+ A novel research on the influence of clients' altruistic behaviors is well explored, which offers deep insights and a significant impact on the domain of fair Federated Learning.

+ A solid Modelling of fair FL systems as altruism coalition formation games (ACFGs) based on game theory and social choice theory.

+ Concrete bounds of optimal egalitarian fairness that an FL coalition can achieve while maintaining core stability under various altruistic behaviors.

+ Novel key insights are obtained: (1) the instability issues emerging from the pursuit of egalitarian fairness are significantly related to the clients’ altruism within the coalition and the configuration of the friends-relationship networks among the clients; (2) the theoretical contributions clarify the quantitative relationships between achievable egalitarian fairness and the disparities in the sizes of local datasets, disproving the misconception that egalitarian fairness inevitably leads to instability.

+ Extensive experiments are conducted to evaluate the consistency of theoretically derived egalitarian fairness bounds with the empirically achieved egalitarian fairness in fair FL settings.

**Weaknesses:**

- In subsections 4.2 and 5.1, it is better to briefly clarify what the colors (blue, red, orange) mean in this context. Do you intend to use colors to better illustrate the results in Table 1?

- In Line 261, though "Proofs for propositions are given in Appendix" was stated, and each proof can be located, it is better to make a clearer pointer for each proposition. For example, which location of the proof in the Appendix is for which proposition?

- In Line 307, Example 1 is used to show which proposition? Better to clarify.

**Questions:**

Please refer to the identified weak points for details.

---

> ### Author Rebuttal · Authors · 2024-08-04
>
> We are delighted that the reviewer recognized the significance of our research and its novel key insights. Thank you for your positive feedback and insightful comments.
> > **W1 (Illustration):**
>
> Thank you for your comment. In Sections 4.2 and 5.1, we utilize color text to better illustrate the results in Table 1. Different colors (blue, red, orange, green) are used to indicate stable coalition structures under different behaviors.
>
> >**W2 (Navigation):**
>
>
> Thank you for your suggestion. We'd like to revise Section 5.2 and connect the propositions to their corresponding proofs in the Appendix to improve clarity and navigation.
>
> > **W3 (Example 1):**
>
> Thank you for your comment.  **Example 1 shows how to calculate the achievable egalitarian fairness bound for heterogeneous behaviors (Lines 303-306)**. To improve clarity, we'd like to revise this paragraph (Lines 303-306) by adding "An example to calculate the achievable egalitarian fairness bound heterogeneous behaviors is as follows:".

---

> ### Comment · Reviewer_LduT · 2024-08-13
> **My comments have been well addressed**
>
> Thank you for the authors' response. My comments have been well addressed.

---

### Author Rebuttal · Authors · 2024-08-04

Dear Chairs and Reviewers,

We kindly thank all the reviewers for their time and for providing valuable feedback on our work. We appreciate that reviewers have pointed out that our work is novel (Reviewer LduT, r5t6),  significant (Reviewer LduT, CtFP) with valuable insights (Reviewer LduT, r5t6), interesting and well-motivated (Reviewer X2VX), and our results are convincing with solid analysis (Reviewers LduT, CtFP, r5t6).

In response to the reviews, we ran a series of experiments to show the generality of our findings under weighted utilitarian welfare function  (Reviewer  CtFP),  network topology  (Reviewer r5t6), number of agents (Reviewer r5t6 and Reviewer X2VX).

Kind regards,

The authors

---

### Decision · Program_Chairs · 2024-09-25

**Decision:**

Accept (poster)

**Comment:**

The reviewers all saw merit in the contributions of the paper, with some having concerns about the significance of the results for practical applications, seeing various important aspects underexplored.  This lead to some feeling that the contribution of the paper as currently written is borderline.  The author response included some additional results that move things in a positive direction in this regard, but there is still room for further refinement.